# DCR$^2$-AD: Dynamic Context Routing and Reasoning Multi-Modal Large Language Model for Anomaly Detection

## Abstract

Recent advances in Multimodal Large Language Models (MLLMs) have shifted the anomaly detection paradigm from traditional classification-based approaches toward a novel diagnostic framework based on MLLM-driven question answering. In contrast to conventional architectures characterized by "single-scenario, single-purpose designs", these models use pretraining to attain robust generalization capabilities and provide expert-level diagnostic performance. However, current MLLM-based anomaly detection methods rely predominantly on internalized knowledge of visual defects, which limits their effectiveness in open-domain settings where anomalies demonstrate significant cross-scenario ambiguity. For example, logical anomalies differ fundamentally from common visual defects, and hence cannot be effectively identified using conventional visual defect-based rules. To overcome this limitation, we propose an innovative Dynamic Context Routing and Reasoning model (DCR$^2$-AD), which integrates knowledge-routed reasoning trajectory synthesis (KR-RTS) and knowledge-routed direct preference optimization (KR-DPO) to improve the model's capacity for appropriate external knowledge utilization during reasoning. We first constructed an object-agnostic knowledge base encompassing extensive defect-related knowledge. By substituting knowledge from correct reasoning trajectories with information drawn from incorrect trajectories, we synthesized erroneous reasoning trajectories. Furthermore, we introduce the KR-DPO algorithm, which conditions on the selectively routed knowledge to promote correct reasoning trajectories and suppress incorrect ones, thereby refining the model's ability to identify optimal reasoning pathways. Through extensive experiments, our approach achieves state-of-the-art performance, attaining 83.36% on the comprehensive MMAD benchmark, surpassing the base model by 6.00%, outperforming ordinary humans by 4.67%, and exceeding the previous best method by 1.41%. These significant gains substantiate the efficacy of our proposed framework. Our code and data will be made publicly available upon publication of the paper.

## 1 Introduction

Automated industrial inspection constitutes an indispensable element of modern manufacturing systems and is critical to ensuring efficient quality assurance throughout production processes (Bergmann et al., 2019b;a; Cao et al., 2023; Chen et al., 2022; Huang et al., 2022; Li et al., 2024; Gao, 2024; Jiang et al., 2022). To identify subtle defects, conventional approaches have introduced "one-class" anomaly detection (AD) techniques (Chen et al., 2024; Fučka et al., 2024; Ho et al., 2024; Hou et al., 2021; Lee & Choi, 2024), which model the distribution of normal samples and identify anomalies by quantifying deviations therefrom. Such methods, however, typically require collecting a large number of defect-free samples from specific operational contexts, leading to limited transferability. CLIP-based AD algorithms leverage pre-trained vision-language models to achieve few-shot anomaly detection in general settings. Nevertheless, these methods are still confined to classification and segmentation tasks, unlike human experts who perform diagnostic analysis based on procedural evidence, thereby restricting their applicability to complex anomaly scenarios encountered in real-world industrial environments. The recent advancement of Multimodal Large Language

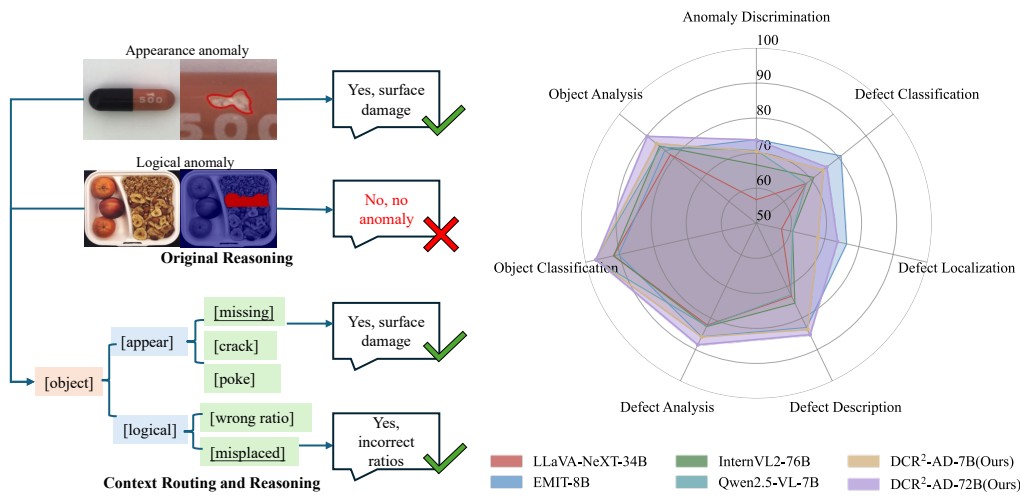

Figure 1: Left: Comparison between original reasoning and our knowledge-routed reasoning in anomaly detection tasks. Our approach can route the reasoning to the correct path, improving the model's performance in specific contexts, such as logical anomaly detection. Right: Our approach achieves the state-of-the-art performance on the MMAD benchmark (with the biggest average value).

Models (MLLMs), endowed with strong universal pre-training and inference capabilities, offers a promising pathway toward developing general-purpose expert-like anomaly detection systems.

Recent advances in Multimodal Large Language Models (MLLMs) have spurred significant innovation in anomaly detection methodologies. For instance, MMAD (Jiang et al., 2024) proposed a comprehensive evaluation benchmark that comprehensively examines the model's baseline capabilities across seven dimensions, including defect detection, defect classification, object detection, and more, becoming an important benchmark for general expert anomaly detection models. Anomaly-OV was the first to apply MLLMs to zero-shot anomaly detection scenarios, demonstrating strong generalization capabilities. Most other methods focus on improving model reasoning capabilities, such as Anomaly-R1 (Chao et al., 2025), EMIT (Guan et al., 2025), IAD-R1 (Li et al., 2025), LR-IAD (Zeng et al., 2025), and OmniAD (Zhao et al., 2025). These methods use the GRPO (Shao et al., 2024) method to enhance the high-quality chain-of-thought (CoT) (Wei et al., 2023) generation capability during model reasoning. However, these methods focus on improving the model's internal capabilities and ignore the model's use of external context, which is crucial for a highly scenario-specific problem like anomaly detection.

Why is external context so critical for expert anomaly detection models? External context is essential for expert anomaly detection models because of the inherent contextual ambiguity of anomalies. Specifically, the semantic meaning of an anomaly varies greatly and depends on the application. For instance, as shown in Figure 1, appearance-based anomalies such as surface scratches or coating defects can often be detected using perceptual features derived from deep learning or image processing. In comparison, logical anomalies such as misassembly, positional errors, or functional failures require reasoning based on structured domain knowledge and contextual constraints, for example, assembly rules, spatial relations, or process sequences. Without integrating such external context, models may be unable to interpret semantically complex anomalies, resulting in false positives or false negatives. Therefore, contextual integration is not only advantageous but necessary to achieve robust, generalizable, and interpretable anomaly detection in diverse and dynamic industrial environments.

In this paper, we propose a Dynamic Context Routing and Reasoning framework for Anomaly Detection (DCR$^2$-AD), designed to improve expert anomaly detection models' capacity to utilize ex-

ternal knowledge for reasoning. The overall architecture includes two key components: knowledge-routed reasoning trajectory synthesis (KR-RTS) and knowledge-routed direct preference optimization (KR-DPO). Specifically, in the KR-RTS stage, we first construct an object-agnostic contextual knowledge base that summarizes a broad range of general defect and non-defect decision patterns. This design ensures high transferability of the knowledge base and establishes a foundation for subsequent context routing and reasoning. We then synthesize reasoning trajectories by manually replacing incorrect contexts to generate negative samples for path rejection, enabling the model to learn how to select accurate contextual reasoning paths. Furthermore, we introduce the KR-DPO algorithm to optimize reasoning trajectories under given input and output conditions, encouraging correct reasoning paths and suppressing incorrect ones. Through these three components, we implement an efficient anomaly detection system that achieves state-of-the-art performance on the comprehensive benchmark MMAD. Extensive experiments show that our 7B model attains 80.83%, outperforming the base Qwen2.5-VL-7B model by 8.64%. Our 72B model achieves 83.36%, representing a 6.00% improvement over the base Qwen2.5-VL-72B model, a 4.67% improvement compared to ordinary human performance, and a 1.41% gain over previous state-of-the-art (SOTA). These significant improvements demonstrate the effectiveness and advancement of our method. Our contributions can be summarized as follows:

- We propose a novel MLLM-based anomaly detection framework, DCR$^2$-AD, that highlights the importance of external context for AD tasks, focusing on enhancing the model's ability to leverage external knowledge for reasoning and improving the overall capability of MLLM-AD systems.

- We introduce two key components, KR-RTS and KR-DPO: KR-RTS improves the model's contextual knowledge selection ability through manually synthesized reasoning trajectories, while KR-DPO enhances reasoning performance by reinforcing positive samples and penalizing negative ones.

- Our model achieves state-of-the-art performance on public benchmarks, demonstrating significant improvements over base models and even human experts, which confirms the effectiveness and advancement of our algorithm.

## 2 RELATED WORK

**Industrial Anomaly Detection (IAD).** IAD is a task with wide application value in the field of computer vision, playing an important role in promptly identifying abnormal components and optimizing production processes. The key issue of IAD lies in identifying and locating anomalous areas and conducting diagnostic analysis. Traditional industrial anomaly detection methods discover anomalies by using unsupervised methods such as comparing with memorized normal samples or comparing with neighboring regions of samples, which including methods based on local area similarity analysis (Li et al., 2024), methods based on memory banks (Gao, 2024; Jiang et al., 2022; Wang et al., 2025), and methods that enhance training data by synthesizing anomalies (Zavrtanik et al., 2021), etc. However, these models rely on pre-defined anomaly concepts, which limit their generalization ability in new scenarios.

**Multimodal Large Language Models for Anomaly Detection(MLLM-based AD).** The recent development of multimodal large models has promoted the MLLM-based methods. MMAD (Jiang et al., 2024) defines 7 key subtasks of MLLM in industrial detection, providing a benchmark for evaluating IAD models. AnomalyGPT (Gu et al., 2024) generates training data by simulating abnormal images and creating corresponding text descriptions, then fine-tunes LLMs to directly evaluate abnormal regions. Anomaly-OneVision (Xu et al., 2025) uses a "look-twice" mechanism to automatically select and highlight abnormal visual tokens, effectively improving performance on anomaly reasoning tasks. LogSAD (Zhang et al., 2025) proposes a "training-free" architecture called "thought matching," which coordinates anomaly scores from different LLM detectors through a calibration module, achieving good results in logical and structural anomaly detection. Most other methods focus on improving model reasoning capabilities, such as Anoma- lyR1 (Chao et al., 2025), EMIT (Guan et al., 2025), IAD-R1 (Li et al., 2025), LR-IAD (Zeng et al., 2025), and OmniAD (Zhao et al., 2025). These methods use the GRPO method to enhance the high-quality CoT generation capability during model reasoning.

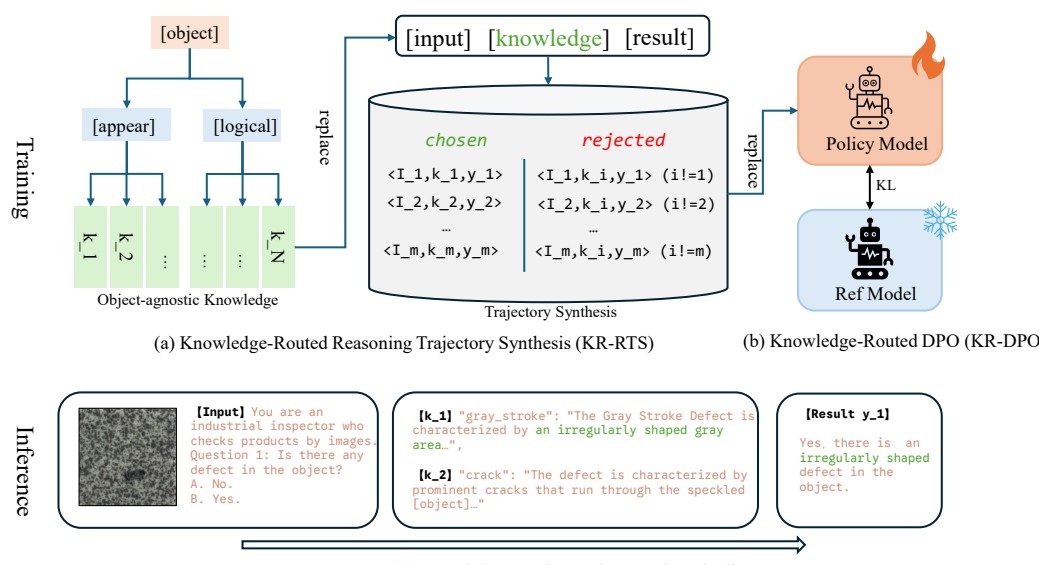

Figure 2: The overall framework of the proposed DCR$^2$-AD. (a) In the process of KR-RTS, we construct the object-agnostic knowledge base according to the collected domain knowledge. Then we synthesize reasoning trajectories by replacing correct knowledge with incorrect ones. (b) We use KR-DPO to optimize the whole framework. (c) The pipeline of inference, green text indicates that external knowledge is correctly routed and utilized.

## 3 METHODS

### 3.1 PRELIMINARY: MLLM-BASED ANOMALY DETECTION VIA LINGUISTIC REASONING

The Multimodal Large Language Model (MLLM) based anomaly detection framework reformulates the visual anomaly detection task as a language-guided, reasoning-intensive visual question answering problem. The core idea is to leverage the powerful language understanding and generation capabilities of MLLMs to fuse visual information with textual instructions and context, thereby conducting step-by-step CoT reasoning to arrive at a final detection decision.

The process begins with multimodal input representation, where an input image $\mathcal{I}$ and a textual query $\mathcal{Q} = (\mathcal{S}, \mathcal{C}_{\text{ext}})$ composed of an instruction component and external context are processed through dedicated tokenizers. The visual tokenizer $\phi_{\text{vis}}$ transforms the image into a sequence of visual tokens $\mathbf{V} = \phi_{\text{vis}}(\mathcal{I}) = [v_1, v_2, \ldots, v_M]$, while the textual tokenizer $\phi_{\text{txt}}$ converts the query into text tokens $\mathbf{T}_q = \phi_{\text{txt}}(\mathcal{Q}) = [t_1, t_2, \ldots, t_N]$. These token sequences are then concatenated to form the unified multimodal input $\mathbf{X} = [\mathbf{V}; \mathbf{T}_q]$ for the MLLM.

The reasoning phase operates through autoregressive generation, where the model parameterized by $\Theta$ produces an output sequence $\mathcal{Y}$ by sequentially predicting each token based on the entire input and previously generated tokens, following the probability distribution $P(y_k | \mathbf{X}, y_{<k}; \Theta)$. Critically, the generation process typically produces not just a direct answer but first generates an internal reasoning chain $\mathcal{C}_{\text{int}}$ that explicates the diagnostic steps, followed by the final anomaly classification $\mathcal{A}$, such that $\mathcal{Y} = [\mathcal{C}_{\text{int}}, \mathcal{A}]$. This internal context $\mathcal{C}_{\text{int}}$ interacts with the external context $\mathcal{C}_{\text{ext}}$ from the input query through the model's attention mechanisms, creating a synergistic effect that enhances both the accuracy and interpretability of the final detection outcome. The overall process is summarized as follows:

$$\underbrace{(\mathcal{I}, \mathcal{S}, \mathcal{C}_{\text{ext}})}_{\text{input}} \xrightarrow{\text{Tokenization}} \underbrace{(\mathbf{V}, \mathbf{T}_Q)}_{\text{tokens}} \xrightarrow{\text{MLLM}_\Theta} \underbrace{(\mathcal{C}_{\text{int}}, \mathcal{A})}_{y} \tag{1}$$

| Key | MVTec.zipper.combined |
|---|---|
| *Knowledge (k_1)* | <[object] Defects>
Description: The examined [object] exhibit multiple defects that collectively affect their functionality and aesthetic integrity. Common issues include:
- **Broken or split teeth:** Sections of the [object] teeth are either missing or misaligned, disrupting the normal interlocking pattern necessary for the [object] to function properly, often located at the center or bottom area of the [object].
- **Fabric border fraying:** Areas along the fabric edges reveal signs of wear, such as frayed or torn fabric, suggesting improper stitching or material degradation. This fraying can threaten the structural integrity of the [object] attachment to the surrounding garment or fabric.
- **Squeezed or misaligned teeth:** Certain sections of the [object] have teeth that appear compressed or out of alignment, which can hinder the smooth operation of the [object] during usage.
- **Fabric interior anomalies:** There are visible irregularities in the fabric interior, such as pulled threads or snags, which can further compromise both the appearance and function of the [object].
Overall, these defects indicate a need for quality control measures, as they could lead to failure during use or an unsatisfactory visual presentation. |

Figure 3: An example of object-agnostic knowledge. Each defect is defined by a detailed expert description. The red text represents the specific category of the object being replaced, ensuring that the knowledge is object-agnostic.

The training paradigm for such models typically involves a two-stage approach. Initially, Supervised Fine-Tuning (SFT) is employed using high-quality annotated data $\mathcal{D} = (\mathbf{X}i, \mathcal{Y}i)$ to optimize the model parameters by maximizing the likelihood of correct outputs:

$$\mathcal{L}_{\text{SFT}}(\Theta) = -\mathbb{E}_{(\mathbf{X},\mathcal{Y})\sim\mathcal{D}} \sum_{k=1}^{L} \log P(y_k|\mathbf{X}, y_{<k}; \Theta) \tag{2}$$

Subsequently, reinforcement learning methods such as GRPO or PPO are applied to further align the model outputs with human preferences, optimizing the objective:

$$\mathcal{L}_{\text{RL}}(\Theta) = \mathbb{E}_{\mathbf{X}\sim\mathcal{D},\mathcal{Y}\sim\pi_\Theta}[R(\mathcal{Y},\mathbf{X})] - \beta \cdot D_{\text{KL}}(\pi_\Theta||\pi_{\text{ref}}) \tag{3}$$

where $R$ is a reward function that evaluates the quality of the output and the KL divergence term ensures that the model does not deviate excessively from the SFT baseline $\pi_{\text{ref}}$.

### 3.2 DYNAMIC CONTEXT ROUTING AND REASONING ANOMALY DETECTION

The overall framework of our proposed DCR$^2$-AD is shown in Figure 2, at the stage of KR-RTS, we first constructed an object-agnostic knowledge base encompassing extensive defect-related knowledge. By substituting knowledge from correct reasoning trajectories with information drawn from incorrect trajectories, we synthesized erroneous reasoning trajectories. Furthermore, we introduce the KR-DPO algorithm, which conditions on the selectively routed knowledge to promote correct reasoning trajectories and suppress incorrect ones, thereby refining the model's ability to identify optimal reasoning pathways.

### 3.3 KNOWLEDGE-ROUTED REASONING TRAJECTORY SYNTHESIS (KR-RTS)

**Object-agnostic Knowledge base Construction** To validate the effectiveness of external knowledge bases, we constructed one for subsequent experiments. We collected domain knowledge from the MMAD dataset and manually constructed and expanded domain knowledge specific to Real-IAD, forming sample pairs of $< defect\_type, knowledge >$. This contains specialized knowledge features for determining each defect/non-defect, covering a variety of industrial quality inspection scenarios. We then performed object-agnostic processing, replacing all words referring to object names with the placeholder "[object]". This approach allows us to extract common information about defects rather than information bound to objects, resulting in greater transferability. After this process, we obtained 147 pieces of object-agnostic domain knowledge, an example is shown in Figure 3.

**Trajectory Synthesis** KR-RTS aims to train the model to establish correct knowledge routing capabilities. The construction process is based on a key principle: performing perfect reasoning under incorrect knowledge premises still leads to erroneous overall responses. The specific construction method is as follows:

Given a training sample $(x, y_w)$, where $x$ contains the query $q$ and the true relevant knowledge source $k_w$, and $y_w$ is the correct response based on $k_w$. We first randomly sample an incorrect

knowledge source $k_l$ from the knowledge base $\mathcal{K}$, satisfying $k_l \neq k_w$ and possessing certain deceptive similarities with the current query in surface features. Subsequently, we instruct annotators to generate a logically consistent reasoning process that conforms to the norms of $k_l$ but is erroneous relative to the actual scenario.

This process can be formally represented as: $y_l^{\text{select}} = f_{\text{generate}}(x, k_l)$, where $f_{\text{generate}}$ maintains the same reasoning rigor and linguistic as the positive sample, with the only difference being the initial knowledge selection. For example, when detecting scratches on an [object] surface, the positive sample selects "Surface Defect Detection Standard" as the knowledge source, while the negative sample may select "Internal Structure Defect Standard" as the incorrect knowledge source, then strictly follows the latter's specifications to analyze the [object] surface image, producing a formally correct but substantively erroneous judgment.

### 3.3.1 Knowledge-Routed Direct Preference Optimization (KR-DPO)

The objective function of standard Direct Preference Optimization (DPO) (Rafailov et al., 2024) utilizes implicit reward modeling to optimize the policy. Given a preference data set $\mathcal{D} = \{(x, y_w, y_l)\}$, where $y_w$ represents the preferred response and $y_l$ represents the dispreferred response, the standard DPO loss function is defined as

$$\mathcal{L}_{\text{DPO}}(\pi_\theta; \pi_{\text{ref}}) = -\mathbb{E}_{(x, y_w, y_l) \sim \mathcal{D}} \left[ \log \sigma \left( \beta \log \frac{\pi_\theta(y_w|x)}{\pi_{\text{ref}}(y_w|x)} - \beta \log \frac{\pi_\theta(y_l|x)}{\pi_{\text{ref}}(y_l|x)} \right) \right] \quad (4)$$

where $\sigma$ denotes the sigmoid function, $\beta$ is the temperature parameter, and $\pi_{\text{ref}}$ represents the reference policy.

Within the KR-DPO framework, we decompose the complete response $y$ into a knowledge $k$ and a knowledge-based reasoning content $c$, such that $y = (k, c)$. The objective of KR-DPO is to ensure the model selects the correct knowledge source $k_w$ rather than an incorrect one $k_l$. We define the path selection optimization objective as:

$$\mathcal{L}_{\text{path}} = -\mathbb{E}_{(x, k_w, k_l, c) \sim \mathcal{D}_{\text{select}}} \left[ \log \sigma \left( \beta \log \frac{\pi_\theta(k_w, c|x)}{\pi_{\text{ref}}(k_w, c|x)} - \beta \log \frac{\pi_\theta(k_l, c|x)}{\pi_{\text{ref}}(k_l, c|x)} \right) \right] \quad (5)$$

where $\mathcal{D}_{\text{select}}$ contains synthesized samples, meaning $k_l \neq k_w$ but the reasoning content $c$ remains logically self-consistent within their respective knowledge sources. This objective compels the model to learn that even with perfect reasoning, an incorrect knowledge premise renders the overall response unacceptable.

KR-DPO models knowledge selection as an integral part of the generation process rather than as a separate classification task. The model explicitly generates the choice of the source of knowledge and its justification.

## 4 Experiments

### 4.1 Datasets

We collected data from Real-IAD (Wang et al., 2024) and constructed the AD-Instruct-10K dataset for supervised fine-tuning, and the AD-KRDPO-1K dataset for KR-DPO training. AD-KRDPO-1K synthesizes the inference path through KR-RTS. For more information about the dataset, please refer to the Appendix A.1.

### 4.2 Evaluation

MMAD (Jiang et al., 2024) is an industrial anomaly detection dataset that uses GPT-4V (Hurst et al., 2024) to generate semantic annotations, questions, and options for testing from publicly available visual datasets. It contains 8,366 samples, seven key subtasks, and a total of 39,673 multiple-choice questions. The seven sub-tasks are: Anomaly Discrimination, Defect Classification, Defect Localization, Defect Description, Defect Analysis, Object Classification, and Object Analysis. During evaluation, the model's performance was compared across two settings: 1-shot, 0-shot. In the 1-shot

setting, in addition to the query image, a random normal image from the dataset is provided to the model, which can use this image as a template to understand the normal state; in the 0-shot setting, no additional reference images are provided.

### 4.3 IMPLEMENTATION DETAILS

For supervised fine-tuning, we employed AdamW (Loshchilov & Hutter, 2019) as the optimizer coupled with CosineAnnealingWarmRestarts (Loshchilov & Hutter, 2017) for learning rate scheduling. We configured the initial learning rate at $1 \times 10^{-5}$ with a warmup ratio of 0.05. The $DCR^2$-AD-7B model was trained on 2 A800 GPUs for a single epoch (requiring approximately 28 hours), maintaining a per-device batch size of 1. For the larger $DCR^2$-AD-72B variant, we implemented LoRA-based fine-tuning across 6 A800 GPUs, also for a single epoch (approximately 60 hours), while maintaining the same per-device batch size of 1. During the knowledge-routed DPO fine-tuning phase, we utilized differential learning rates: $1 \times 10^{-6}$ for the 7B model and $2 \times 10^{-5}$ for the 72B model, requiring 2 and 4 hours of computational time, respectively. For inference on the MMAD benchmark, the 7B model required 12 hours of processing time, while the baseline 72B model necessitated over 40 hours for complete evaluation. By implementing vLLM optimization techniques for the 72B model, we successfully reduced inference time to 26 hours without any statistically significant degradation in performance metrics.

### 4.4 MAIN RESULTS

We conducted a detailed comparison of the current mainstream MLLM models, which can be broadly categorized into two types: closed-source models and open-source models. Among the closed-source models, we selected OpenAI's GPT-4-o/GPT-4o-mini (Achiam et al., 2023), Google's Gemini-1.5-flash/pro (Reid et al., 2024), and Claude-3.5-sonnet (Anthropic). These commercial closed-source models typically exhibit superior model performance, and we evaluated their effectiveness through API calls. For open-source models, we compared a wide range of models, including the LLaVA-NeXT series (Liu et al., 2024), InternVL series (Chen et al., 2023), Deepseek series (Wu et al., 2024), Qwen series (Bai et al., 2025), and other custom models (Gu et al., 2024; Hu et al., 2024; Team, 2025; Team et al., 2025).

As shown in Table 1, among these models, our model achieves the current state-of-the-art (SOTA) performance, achieving an average score of $80.37\%$ across all seven tasks in the MMAD evaluation metric. This surpasses the best closed-source model, GPT-4o, by $5.45\%$ points, and the best open-source model, Qwen2.5-VL-72B-Instruct, by $3.01\%$ points. Upon further examination, it can be observed that MLLM generally performs poorly in subtasks such as anomaly detection and defect localization, with a significant gap compared to human performance. This is primarily because such scenarios place greater demands on the model's visual capabilities. In tasks such as defect classification and object classification; and in the three tasks of defect description, defect analysis, and object analysis, large models leverage the powerful text capabilities of LLM to easily outperform human performance. Our model, through in-depth learning of contextual information, also performs exceptionally well in these tasks. Additionally, to balance both false positives (misdetections) and false negatives (missed detections), we specifically evaluated the F1 score for the Anomaly Discrimination task. Our 7B model achieved the best performance, attaining a score of 78.45% and outperforming GPT-4o by 7.41%. In summary, our $DCR^2$-AD-72B model achieves leading performance in comprehensive benchmarks.

#### 4.4.1 IMPACT OF DOMAIN KNOWLEDGE

MMAD provides domain knowledge to help improve model performance. In this setting, we can explore the capabilities of the model within the boundaries of domain expertise. Table 2 demonstrates that our $DCR^2$-AD-72B has been further improved from $80.37\%$ to $83.36\%$, with this metric more colse to expert human performance, marking a new milestone for MLLM large models in anomaly detection tasks. It can also be concluded that as the model size increases, the gain from domain knowledge diminishes, indicating that larger models inherently possess richer domain knowledge, thereby reducing reliance on external knowledge.

Table 1: Performance comparison in MMAD with the standard 1-shot setting. Bold type indicates the best performance, underlined type indicates the second best performance.

| Model | Scale | F1 | Anomaly | Defect | | | | Object | | Average |
|---|---|---|---|---|---|---|---|---|---|---|
| | | | Dis. | Cls. | Loc. | Des. | Ana. | Cls. | Ana. | |
| Random Chance | - | - | 50.00 | 25.00 | 25.00 | 25.00 | 25.00 | 25.00 | 25.00 | 28.57 |
| Human (expert) | - | - | 95.24 | 75.00 | 92.31 | 83.33 | 94.20 | 86.11 | 80.37 | 86.65 |
| Human (ordinary) | - | - | 86.90 | 66.25 | 85.58 | 71.25 | 81.52 | 89.58 | 69.72 | 78.69 |
| Claude-3.5-sonnet | - | 41.92 | 60.14 | 60.14 | 48.81 | 67.13 | 79.11 | 85.19 | 79.83 | 68.36 |
| Gemini-1.5-flash | - | 72.40 | 58.58 | 54.70 | 49.10 | 66.53 | 82.24 | 91.47 | 79.71 | 68.90 |
| Gemini-1.5-pro | - | 57.60 | 68.63 | 60.12 | 58.56 | 70.38 | 82.46 | 89.20 | 82.25 | 73.09 |
| GPT-4o-mini | - | 68.67 | 64.33 | 48.58 | 38.75 | 63.68 | 80.40 | 88.56 | 79.74 | 66.29 |
| GPT-4o | - | 71.04 | 68.63 | **65.80** | 55.62 | 73.21 | 83.41 | **94.98** | 82.80 | 74.92 |
| AnomalyGPT | 7B | 76.68 | 65.57 | 27.49 | 27.97 | 36.86 | 32.11 | 29.84 | 35.82 | 36.52 |
| InternLM-XComposer2-VL | 7B | 27.16 | 55.85 | 41.80 | 48.27 | 57.52 | 76.60 | 74.34 | 77.75 | 61.73 |
| LLaVA-OneVision | 7B | 9.10 | 51.77 | 46.13 | 41.85 | 62.19 | 69.73 | 90.31 | 80.93 | 63.27 |
| MiniCPM-V2.6 | 8B | 45.31 | 57.31 | 49.22 | 43.28 | 65.86 | 75.24 | 92.02 | 80.80 | 66.25 |
| InternVL3 | 8B | 41.23 | 68.69 | 55.09 | 59.98 | 70.83 | 80.58 | 87.24 | 82.92 | 72.19 |
| Keye-VL-Preview | 8B | 58.41 | 65.25 | 45.34 | 43.53 | 56.56 | 40.57 | 91.59 | 77.81 | 58.66 |
| Deepseek-vl2-small | 2.8B | 38.94 | 62.84 | 49.69 | 44.86 | 65.47 | 78.00 | 92.42 | 81.47 | 67.82 |
| Qwen2.5-VL-Instruct | 7B | 72.49 | 71.10 | 56.02 | 60.69 | 64.13 | 78.26 | 91.49 | 83.67 | 72.19 |
| MiMo-VL-SFT | 7B | 72.77 | 58.26 | 62.17 | 59.40 | 72.64 | 84.50 | 90.02 | 83.19 | 72.88 |
| MiMo-VL-RL | 7B | 73.85 | 56.60 | 63.03 | 61.89 | 72.60 | 83.68 | 90.35 | 82.86 | 73.00 |
| MiMo-VL-SFT(SFT) | 7B | 73.52 | 72.32 | 60.61 | 64.57 | 78.42 | 84.98 | 91.05 | 83.56 | 76.50 |
| MiMo-VL-RL(SFT) | 7B | 74.30 | 69.89 | 60.20 | 70.36 | 78.11 | 84.28 | 89.29 | 83.33 | 76.49 |
| Qwen2.5-VL-7B-Instruct(SFT) | 7B | 72.66 | 73.27 | 56.79 | 63.40 | 72.45 | 83.20 | 92.91 | 86.88 | 75.56 |
| **DCR$^2$-AD-7B(Ours)** | 7B | **78.45** | 70.00 | 62.06 | 69.60 | 79.34 | 84.75 | 93.52 | 86.79 | 78.01 |
| Qwen2.5-VL-Instruct | 32B | 68.19 | 70.88 | 58.29 | 62.40 | 65.38 | 83.16 | 76.89 | 84.91 | 71.70 |
| LLaVA-NeXT | 34B | 54.44 | 57.92 | 48.79 | 52.87 | 71.34 | 80.28 | 81.12 | 77.80 | 67.16 |
| Qwen2.5-VL-Instruct | 72B | 74.08 | 72.96 | 62.71 | 68.58 | 74.65 | 82.12 | 94.41 | 86.12 | 77.36 |
| InternVL2 | 76B | 64.40 | 68.25 | 54.22 | 56.66 | 66.30 | 80.47 | 86.40 | 82.92 | 70.75 |
| Qwen2.5-VL-72B-Instruct(SFT) | 72B | 76.25 | **74.60** | 64.15 | **72.40** | 81.16 | 86.18 | 93.92 | 88.45 | 80.12 |
| **DCR$^2$-AD-72B(Ours)** | 72B | 75.59 | 74.37 | 64.60 | 71.71 | **82.23** | **87.17** | 93.24 | **89.28** | **80.37** |

Table 2: Performance comparison in MMAD with domain knowledge under the 1-shot setting. Bold type indicates the best performance, underlined type indicates the second best performance.

| Model | Scale | Anomaly | Defect | | | | Object | | Average | Improv. |
|---|---|---|---|---|---|---|---|---|---|---|
| | | Dis. | Cls. | Loc. | Des. | Ana. | Cls. | Ana. | | |
| Human (expert) | - | 95.24 | 75.00 | 92.31 | 83.33 | 94.20 | 86.11 | 80.37 | 86.65 | - |
| Human (ordinary) | - | 86.90 | 66.25 | 85.58 | 71.25 | 81.52 | 89.58 | 69.72 | 78.69 | - |
| Qwen2.5-VL-Instruct | 7B | 70.74 | 68.11 | 60.58 | 72.64 | 82.59 | **97.09** | 84.79 | 76.65 | +4.46 |
| MiMo-VL-SFT | 7B | 69.1 | 71.27 | 77.93 | 85.08 | 96.64 | 85.39 | 78.46 | +5.58 | |
| MiMo-VL-RL | 7B | 66.87 | 71.93 | 65.65 | 77.19 | 85.91 | 96.26 | 84.30 | 78.30 | +5.30 |
| MiMo-VL-SFT(SFT) | 7B | **74.95** | 74.63 | 69.98 | 82.89 | 87.01 | 96.40 | 85.53 | 81.63 | +5.13 |
| MiMo-VL-RL(SFT) | 7B | 70.39 | 72.97 | 72.85 | 82.90 | 86.63 | 96.05 | 84.61 | 80.91 | +4.42 |
| Qwen2.5-VL-7B-Instruct(SFT) | 7B | 71.38 | 71.81 | 63.89 | 78.55 | 83.99 | 97.05 | 86.11 | 78.97 | +3.41 |
| **DCR$^2$-AD-7B(Ours)** | 7B | 70.46 | 74.28 | 68.07 | 78.55 | 86.01 | 96.9 | 86.40 | 80.83 | +2.82 |
| InternVL2 | 26B | 68.64 | 67.32 | 53.81 | 70.84 | 82.18 | 93.81 | 83.31 | 74.27 | +5.66 |
| LLaVA-NeXT | 34B | 56.72 | 68.22 | 57.36 | 73.12 | 82.24 | 91.78 | 81.35 | 72.97 | +5.81 |
| InternVL2 | 40B | 70.01 | 70.09 | 56.89 | 73.29 | 83.26 | 96.50 | 84.41 | 76.35 | **+6.75** |
| Qwen2.5-VL-Instruct | 72B | 71.63 | 73.85 | 69.11 | 79.18 | 83.87 | 96.81 | 87.04 | 80.21 | +2.85 |
| InternVL2 | 76B | 66.68 | 70.95 | 60.57 | 75.32 | 82.71 | 91.71 | 85.29 | 76.18 | +5.43 |
| Qwen2.5-VL-72B-Instruct(SFT) | 72B | 73.10 | **76.01** | 73.68 | 85.01 | 87.87 | 96.74 | 88.77 | 83.02 | +2.90 |
| **DCR$^2$-AD-72B(Ours)** | 72B | 73.77 | 75.65 | **73.72** | **85.27** | **88.47** | 96.85 | **89.76** | **83.36** | +2.99 |

### 4.4.2 ZERO-SHOT AD PERFORMANCE

Under zero-shot setting, the model has no reference to normal samples without defects, which further tests the model's ability to reason about visual features and internal knowledge. As shown in Table 3, we also achieved the best results in the MMAD zero-shot setting, 78.60%. This indicates that the model itself has strong anomaly detection capabilities and can generalize to new scenarios without

Table 3: Performance comparison in MMAD under the 0-shot setting. Bold type indicates the best performance, underlined type indicates the second best performance.

| Model | Scale | Anomaly | Defect | | | | Object | | Average |
|---|---|---|---|---|---|---|---|---|---|
| | | Dis. | Cls. | Loc. | Des. | Ana. | Cls. | Ana. | |
| Gemini-1.5-flash | - | 58.43 | 49.93 | 53.11 | 63.07 | 82.83 | - | - | 68.58 |
| LLaVA-NEXT-Interleave | 7B | 58.39 | 36.98 | 48.98 | 51.51 | 66.64 | - | - | 60.04 |
| InternLM-XComposer2-VL | 7B | 58.33 | 43.10 | 54.56 | 57.84 | 75.30 | - | - | 62.78 |
| Cambrian | 8B | 55.60 | 32.53 | 35.39 | 43.46 | 49.14 | 78.15 | 67.22 | 51.64 |
| InternVL3 | 8B | 64.67 | 50.91 | 60.06 | 65.49 | 77.16 | 82.01 | 83.58 | 69.13 |
| Keye-VL-Preview | 8B | 61.85 | 51.37 | 52.94 | 63.79 | 43.99 | 85.05 | 80.10 | 62.73 |
| Deepseek-vl2-tiny | 1B | 50.60 | 43.21 | 54.97 | 65.52 | 75.66 | 88.64 | 76.88 | 65.07 |
| Deepseek-vl2-small | 2.8B | 62.33 | 49.69 | 55.16 | 67.84 | 79.14 | 93.01 | 83.09 | 70.04 |
| Qwen2.5-VL-Instruct | 7B | 60.45 | 50.16 | 57.80 | 60.73 | 75.31 | 93.24 | 84.96 | 68.95 |
| MiMo-VL-SFT | 7B | 64.80 | 55.90 | 54.52 | 68.29 | 83.39 | 91.29 | 84.87 | 71.87 |
| MiMo-VL-RL | 7B | 66.06 | 57.78 | 56.95 | 69.48 | 83.72 | 91.63 | 84.97 | 72.94 |
| MiMo-VL-SFT(SFT) | 7B | 67.59 | 58.06 | 61.99 | 76.10 | 84.51 | 91.52 | 83.28 | 74.72 |
| MiMo-VL-RL(SFT) | 7B | 67.27 | 58.26 | 69.13 | 75.83 | 84.10 | 90.08 | 83.49 | 75.45 |
| Qwen2.5-VL-7B-Instruct(SFT) | 7B | 65.55 | 51.44 | 63.03 | 68.72 | 81.49 | 92.64 | 85.97 | 72.69 |
| **DCR$^2$-AD-7B(Ours)** | 7B | 66.2 | 60.77 | 68.45 | 77.52 | 83.97 | **94.42** | 86.05 | 76.77 |
| LLaVA-NeXT | 34B | 60.25 | 51.57 | 55.49 | 71.62 | 80.43 | - | - | 68.45 |
| Qwen2.5-VL-Instruct | 72B | 66.22 | 57.99 | 62.95 | 72.21 | 81.01 | 93.92 | 86.51 | 74.40 |
| InternVL2 | 76B | 64.30 | 51.19 | 54.20 | 63.46 | 79.92 | 89.34 | 83.48 | 69.41 |
| Qwen2.5-VL-72B-Instruct(SFT) | 72B | **69.23** | **62.82** | **69.77** | 79.55 | 86.34 | **94.10** | 88.39 | **78.60** |
| **DCR$^2$-AD-72B(Ours)** | 72B | 68.71 | 62.13 | 69.23 | **80.16** | **86.37** | 93.07 | **89.23** | 78.42 |

samples. This is also a shortcoming that traditional unsupervised anomaly detection methods cannot achieve.

## 4.5 ABLATION STUDY

We validate the effectiveness of KR-DPO through an ablation experiment on Qwen2.5-VL-7B-Instruct, with results presented in Table 4. The baseline model, fine-tuned only with SFT, achieves an average score of 75.56%. By incorporating our KR-DPO stage, the full model's performance improved on nearly all subtasks, with the notable exception of Anomaly Discrimination, boosting

Figure 4: Ablation study about KR-DPO.

| Model | Anomaly | Defect | | | | Average |
|---|---|---|---|---|---|---|
| | Dis. | Cls. | Loc. | Des. | Ana. | |
| **Full model** | 70.00 | **62.06** | **69.6** | **79.34** | **84.75** | **78.01** |
| w/o. KR-DPO | **73.27** | 56.79 | 63.40 | 72.45 | 83.20 | 75.56 |
| w/o. SFT | 71.10 | 56.02 | 60.69 | 64.13 | 78.26 | 72.19 |

the average performance to 78.01%. This significant improvement demonstrates that KR-DPO plays a crucial role in aligning the model with desired behaviors and is a key contributor to its final performance.

## 5 CONCLUSIONS

In this study, we introduced the Dynamic Context Routing and Reasoning model for Anomaly Detection (DCR²-AD), a novel framework designed to overcome the limitations of existing MLLM-based approaches in open-domain anomaly detection. By integrating knowledge-routed reasoning path synthesis (KR-RPS) and knowledge-routed direct preference optimization (KR-DPO), the model effectively leverages external knowledge to enhance reasoning accuracy and adaptability across diverse anomaly types. Extensive experimental results on the MMAD benchmark demonstrate that our method achieves state-of-the-art performance with an accuracy of 83.36%, outperforming both baseline models and ordinary humans, thereby validating the efficacy of the proposed approach. For future work, we plan to extend the knowledge base to include more domains and explore real-time adaptive routing mechanisms.

## 6 ETHICS STATEMENT

This work adheres to the ICLR Code of Ethics. In this study, no human subjects or animal experimentation was involved. All datasets used, including AD-Instruct-10K and AD-KRDPO-1K, were sourced in compliance with relevant usage guidelines, ensuring no violation of privacy. We have taken care to avoid any biases or discriminatory outcomes in our research process. No personally identifiable information was used, and no experiments were conducted that could raise privacy or security concerns. We are committed to maintaining transparency and integrity throughout the research process.

## 7 REPRODUCIBILITY STATEMENT

We have made every effort to ensure that the results presented in this paper are reproducible. All code and datasets have been made publicly available in an anonymous repository to facilitate replication and verification. The experimental setup, including training steps, model configurations, and hardware details, is described in detail in the paper. We have also provided a full description of $DCR^2-AD$, to assist others in reproducing our experiments.

Additionally, public anomaly detection datasets, such as MMAD, Real-IAD, are publicly available, ensuring consistent and reproducible evaluation results.

We believe these measures will enable other researchers to reproduce our work and further advance the field.

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

# A APPENDIX

## A.1 AD-INSTRUCT-10K

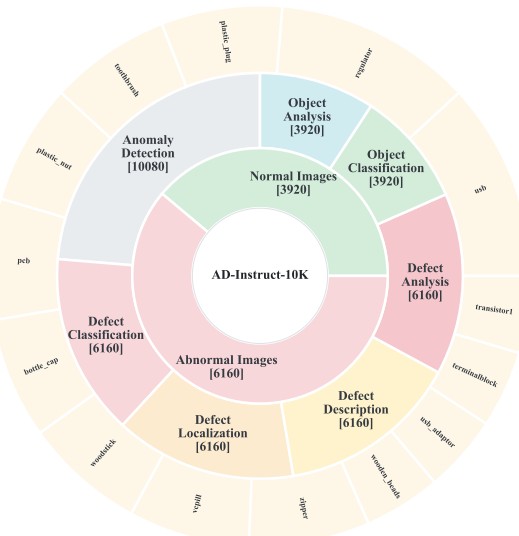

Figure 5: Statistical overview of AD-Instruct-10K. The innermost layer represents image components, the middle layer depicts subtask composition, and the outermost layer illustrates object categories.

Existing industrial anomaly detection datasets (Bergmann et al., 2019b; Wang et al., 2024) primarily emphasize object class labels, visual mask annotations, and defect class labels. However, they often lack contextual annotations, limiting their utility for multimodal understanding. To address this limitation, we introduce AD-Instruct-10K, a meticulously curated instruction-tuning dataset specifically designed for industrial visual anomaly detection. AD-Instruct-10K not only preserves the authentic characteristics of industrial anomalies but also supplements them with precisely localized, domain-specific contextual annotations that capture the subtle nuances of various manufacturing defects across diverse industrial sectors.

Our AD-Instruct-10K represents a significant advancement through its novel construction methodology—a multi-stage generation pipeline that systematically transforms and enriches real-world industrial defect images from the Real-IAD dataset (Wang et al., 2024). For initial QA generation, we implement a sophisticated templated question-answer (QA) generation framework and leverage GPT-4o with carefully engineered prompts that include explicit instructions. Next, we employ a strategic randomization process for answer options, deliberately shuffling both the answer designators (A, B, C, D) and their corresponding textual content. This crucial step ensures that models trained on our dataset learn the substantive content of anomaly descriptions rather than memorizing

positional patterns of correct answers. For quality assurance, we implement a secondary verification mechanism utilizing the pixel-precise segmentation masks available in the Real-IAD dataset to validate the spatial accuracy of GPT-4o's defect localization descriptions, thereby ensuring annotation fidelity to the actual anomaly boundaries.

In the end, there are five distinct types of question-answer pairs generated for each image, closely following MMAD's established paradigm. AD-Instruct-10K thus presents a comprehensive and extensible resource for advancing the state-of-the-art in visual anomaly detection within the field of industrial inspection and quality control.

## A.2 AD-KRDPO-1K

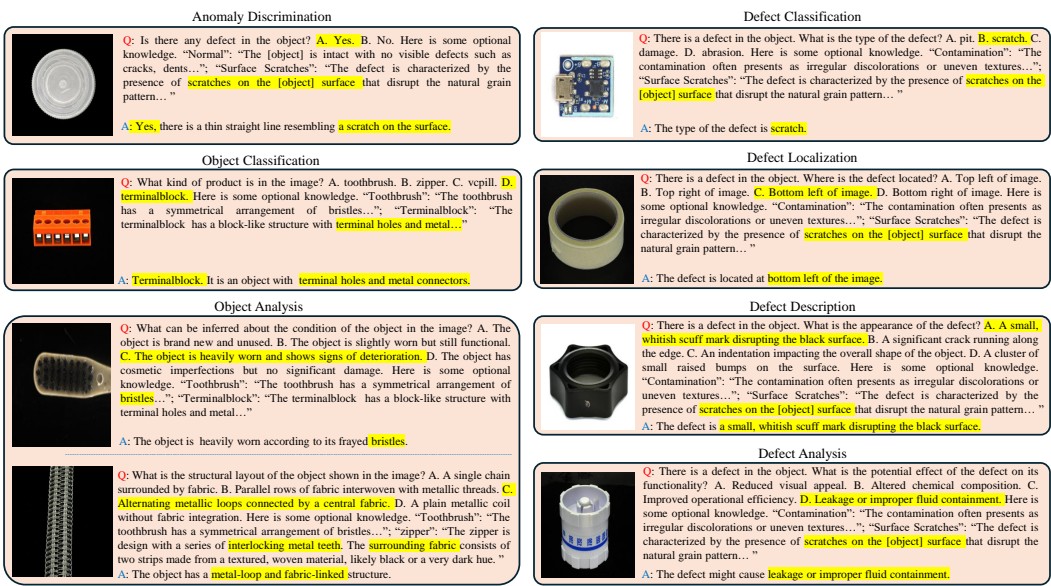

Figure 6: Examples of 7 subtasks of AD-KRDPO-1K. Each question is presented in a multiple-choice format and includes several distractor options. We present different categories of objects in various examples to demonstrate the diversity.

Building upon AD-Instruct-10K, we developed a systematic approach to extract challenging samples for Direct Preference Optimization (DPO) fine-tuning. We first employed our model previously fine-tuned on AD-Instruct-10K to generate predictions across the entire dataset. We then implemented a precise filtering criterion, preserving only samples where the probability assigned to the ground-truth answer exhibited marginal deviation from random chance performance. Specifically, we retained two categories of samples: those with probabilities slightly exceeding random guessing (indicating correct but uncertain predictions) and those with probabilities marginally below random guessing (representing incorrect but potentially recoverable predictions). To operationalize this selection mechanism, we established a quantitative threshold of 0.1 for defining "marginal deviation." This rigorous filtration process resulted in 1K chosen instruct data, a concentrated subset of boundary cases ideally suited for preference-based optimization.

Further, we augmented each question-answer pair with structured knowledge elements to create AD-KRDPO-1K, a dataset specifically engineered for knowledge-guided reasoning. For each sample, we systematically incorporated multiple knowledge selections—comprising precisely one correct knowledge path and several deliberately crafted erroneous alternatives. This resulted in a sophisticated structure where each sample contains one or more <knowledge><answer> pairs, creating a controlled experimental environment for discriminative learning. The primary objective of AD-KRDPO-1K is to facilitate the development of robust knowledge routing capabilities, enabling models to accurately identify, select, and apply relevant knowledge fragments while systematically

rejecting misleading or irrelevant information paths. This approach represents a significant advancement in training visual reasoning systems that can transparently justify their decisions through explicit knowledge utilization.

# B    LLM USAGE

Large Language Models (LLMs) were used to aid in the writing and polishing of the manuscript. Specifically, we used an LLM to assist in refining the language, improving readability, and ensuring clarity in various sections of the paper. The model helped with tasks such as sentence rephrasing, grammar checking, and enhancing the overall flow of the text.

It is important to note that the LLM was not involved in the ideation, research methodology, or experimental design. All research concepts, ideas, and analyses were developed and conducted by the authors. The contributions of the LLM were solely focused on improving the linguistic quality of the paper, with no involvement in the scientific content or data analysis.

The authors take full responsibility for the content of the manuscript, including any text generated or polished by the LLM. We have ensured that the LLM-generated text adheres to ethical guidelines and does not contribute to plagiarism or scientific misconduct.

# C    RESPONSE

## C.1    OBJECT-AGNOSTIC DESIGN

We would like to clarify that "object-agnostic" is an intentional, core design choice motivated by the goals of transferability and generalization. Traditional methods emphasize the inherent object-defect relationship, which can improve fitting capability for in-domain data, but may sacrifice generalization capability for out-of-domain objects. We will explain it from the perspectives of conditional probability and causal inference. Traditional methods can be modeled as follows:

$$P(\text{Defect}) = \sum_{\text{obj}} P(\text{obj}) P(\text{Defect} \mid \text{obj})$$

Through training, the model learns to model the posterior distribution $P(\text{Defect} \mid \text{obj})$, which is the "object-defect" latent relationship. However, this approach suffesr from two problems in transfer and generalization. Our method does not ignore the inherent object-defect link but rather encourages deeper feature-defect connections. According to causal inference theory, anomaly detection can be viewed as the reasoning process Input $\rightarrow$ Feature $\rightarrow$ Object $\rightarrow$ Defect. However, over-reinforcing the Object leads to the two aforementioned problems. Therefore, we adopt a causal intervention approach by hiding $obj$ and weakening $P(\text{Defect} \mid \text{obj})$, encouraging the model to focus more on the underlying "feature-defect" connection, i.e., $P(\text{Defect} \mid \text{Feature})$. Features are lower-level than objects, inherently possessing stronger transferability and generalization.

We conducted experiments to demonstrate the effect of the object-agnostic design. Results are shown in the Table 4. We compare four variants: SFT-only, SFT+DPO, SFT+KR-DPO (obj-related), and SFT+KR-DPO (obj-agnostic). Compared to vanilla DPO, KR-DPO with object-specific design causes a 0.97% performance drop, indicating that object-specific KR-DPO leads to degraded performance. In contrast, KR-DPO with object-agnostic design further improves performance by 0.99%, achieving the best results.

## C.2    VERIFICATION FOR LOGICAL ANOMALY DETECTION

we conducted a new subset experiment. We separated "logical anomalies" (e.g., misalignment, proportion errors) and "visual anomalies" (e.g., scratches) from the MMAD, totaling three subsets. The logical subset and structural subset are split from MVTec-LOCO in MMAD. The rest of MMAD is the visual subset. We then evaluated the model's performance on each subset separately. Here, we selected only defective samples and used Anomaly Discrimination accuracy as the metric. Specific results are shown in the Table 5

Additionally, we have added visualizations to demonstrate the consistency between the model's reasoning process (internal chain of thought) and final detection results, as shown at Figure 7. These

Table 4: Ablation study about DPO on Qwen2.5-VL-7B-Instruct in MMAD with the standard 1-shot setting. Bold type indicates the best performance.

| SFT | DPO | KR-DPO(obj-related) | KR-DPO(obj-agnostic) | Anomaly | Defect | | | | | Object | | Average |
|---|---|---|---|---|---|---|---|---|---|---|---|---|
| | | | | Dis. | Cls. | Loc. | Des. | Ana. | Cls. | Ana. | |
| ✓ | | | | **73.27** | 56.79 | 63.40 | 72.45 | 83.20 | 92.91 | **86.88** | 75.56 |
| ✓ | ✓ | | | 70.23 | 62.03 | 69.50 | 79.20 | **84.77** | 93.41 | 86.83 | 77.99 |
| ✓ | | ✓ | | 67.78 | 61.31 | 67.02 | 78.74 | 84.59 | 93.04 | 86.66 | 77.02 |
| ✓ | | | ✓ | 70.00 | **62.06** | **69.60** | **79.34** | 84.75 | **93.52** | 86.79 | **78.01** |

Table 5: Accuracy about different anomalies in MMAD under the 1-shot setting. Bold type indicates the best performance.

| Model | Logical anomalies | Structural anomalies | Visual anomalies |
|---|---|---|---|
| Qwen2.5-VL-7B-Instruct | 30.30 | 59.63 | 46.59 |
| **DCR$^2$-AD-7B(Ours)** | **62.39 (+32.09)** | **82.60 (+22.97)** | **63.97 (+17.38)** |

correct knowledge reasoning paths can enhance the model's reasoning performance and also demonstrate the consistency between the reasoning process and the reasoning results.

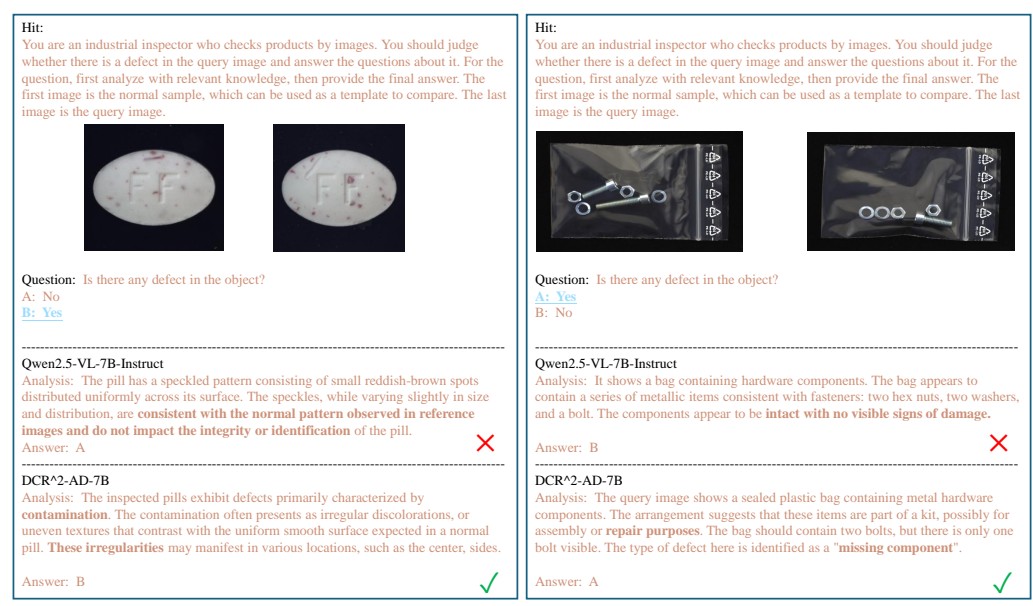

Figure 7: Examples of different anomalies. The left side of the image shows visual anomalies, while the right side shows logical anomalies. Through our comparison, we can observe that our model's reasoning process can capture key discriminative paths for defects, such as irregular discolorations, uneven textures, and digital logic issues.

## C.3 RELATIONSHIP BETWEEN MODEL SCALE AND EXTERNAL KNOWLEDGE UTILITY.

To further investigate the impact of model scale, we aligned experiments using qwen2.5-vl as the base model and added relevant experiments on a smaller-scale 3B model. The results are shown in Table 6. As model size increases, the gains from external knowledge decrease. We believe that

large models (e.g., 72B) have "internalized" more knowledge through their massive parameters, thus reducing their dependence on external knowledge (i.e., a performance ceiling effect). This suggests that DCR$^2$-AD is a key technology that enables small to medium-sized, parameter-efficient models (e.g., 7B) to achieve or approach the performance of large models through effective utilization of external knowledge.

Table 6: Performance gain on different model-scales in MMAD under the 1-shot setting. Bold type indicates the best performance.

| Model | Method | Average(1-shot) | Average(1-shot+DomainKnowledge) | Improv. |
|---|---|---|---|---|
| | Base | 68.87 | 73.95 | **+5.08** |
| Qwen2.5-VL-3B-Instruct | +SFT | 72.56 | 77.51 | +4.95 |
| | +Ours | **74.13** | **78.96** | +4.83 |
| | Base | 72.19 | 76.65 | **+4.46** |
| Qwen2.5-VL-7B-Instruct | +SFT | 75.56 | 78.97 | +3.41 |
| | +Ours | **78.01** | **80.83** | +2.82 |
| | Base | 77.36 | 80.21 | +2.85 |
| Qwen2.5-VL-72B-Instruct | +SFT | 80.12 | 83.02 | +2.90 |
| | +Ours | **80.37** | **83.36** | **+2.99** |

## C.4 COMPARISON WITH COMPETING METHODS LEVERAGING EXTERNAL KNOWLEDGE

To validate the ability of our approach to leverage external knowledge, we select several related methods for compare. All results are evaluated on the MMAD benchmark and the overall scores are shown in Table 7. Compared with other RAG-style baselines, our method achieves the best performance, improving upon the base model by +5.82%. This is because existing RAG methods rely on static retrieval to obtain external knowledge, whereas our framework actively selects the most appropriate knowledge through a dynamic routing mechanism, further enhancing the model's capacity to exploit external information.

Table 7: Performance comparison with other methods which leverages external knowledge on Qwen2.5-VL-7B-Instruct. Our model is in the standard 1-shot setting. Bold type indicates the best performance.

| Model | RAG | Average |
|---|---|---|
| Qwen2.5-VL-7B-Instruct | None | 72.19 |
| Qwen2.5-VL-7B-Instruct+DomainKnowledge | Yes | 76.65 |
| ECHO | Yes | 77.31 |
| GraphRAG | Yes | 77.24 |
| **DCR$^2$-AD-7B(Ours)** | None | **78.01** |

## C.5 COMPARISON WITH OTHER FINE-TUNING STRATEGIES

We compare our approach with several popular fine-tuning recipes, including KTO and GRPO. Results in Table 8 show that the gain brought by DCR$^2$-AD (SFT+KR-DPO) is +2.45%, higher than SFT+GRPO (+2.29%) and SFT+KTO (+1.49%). GRPO relies only on outcome reward, whereas our approach provides dense, step-level reward on the knowledge-selection path, which our experiments show to be more effective.

## C.6 GENERALISATION EVIDENCE

We evaluate on the completely new SSGD dataset, whose objects never appear in either the training set or the knowledge base. Experimental results in Table 9 show that, compared with the baseline, our DCR$^2$-AD-7B achieves 79.12% anomaly-discrimination accuracy on the out-of-domain defect

Table 8: Ablation study about finetuning on Qwen2.5-VL-7B-Instruct in MMAD with the standard 1-shot setting. Bold type indicates the best performance.

| Method | Anomaly | Defect | | | | Object | | Average |
|---|---|---|---|---|---|---|---|---|
| | Dis. | Cls. | Loc. | Des. | Ana. | Cls. | Ana. | |
| Base | 71.10 | 56.02 | 60.69 | 64.13 | 78.26 | 91.49 | 83.67 | 72.19 |
| +SFT | **73.27** | 56.79 | 63.40 | 72.45 | 83.20 | 92.91 | 86.88 | 75.56 |
| +SFT+KTO | 68.37 | 61.07 | 66.90 | 78.53 | **84.79** | 93.13 | 86.56 | 77.05 |
| +SFT+GRPO | 72.52 | 62.03 | 66.12 | 78.69 | 83.95 | **94.46** | **87.16** | 77.85 |
| +SFT+KR-DPO(Ours) | 70.00 | **62.06** | **69.60** | **79.34** | 84.75 | 93.52 | 86.79 | **78.01** |

dataset SSGD, relative to Qwen2.5-VL-7B, and delivers a 17% higher F1 score, demonstrating the strong generalization capability of our object-agnostic design.

Table 9: Performance comparison on SSGD in the 0-shot setting. Bold type indicates the best performance.

| Model | Anomaly Dis. | F1 | Acc.Normal | Acc.Abnormal |
|---|---|---|---|---|
| Qwen2.5-VL-7B-Instruct | 76.67 | 72.79 | **94.85** | 58.50 |
| **DCR$^2$-AD-7B(Ours)** | **79.12** | **89.79** | 64.07 | **94.16** |

## C.7 COMPARISON WITH THE STATE-OF-THE-ART CLOSED-SOURCE MODELS

As the Table 10 shown below, across the three settings of 0-shot, 1-shot and 1-shot (with domain knowledge), our DCR$^2$-AD-72B consistently achieved the best Average performance. Specifically, it outperformed Gemini-2.5-pro by **4.21%** in the 1-shot setting and by **3.40%** in the 0-shot setting, ultimately attaining a state-of-the-art (SOTA) result of **83.36%** in the 1-shot+domain knowledge setting. Notably, among the seven subtasks, our model demonstrated outstanding performance on defect-related tasks, such as defect localization and defect description.

Table 10: . Comparison with the state-of-the-art closed-source models. Bold type indicates the best performance.

| Setting | Model | Anomaly | Defect | | | | Object | | Average |
|---|---|---|---|---|---|---|---|---|---|
| | | Dis. | Cls. | Loc. | Des. | Ana. | Cls. | Ana. | |
| 0-shot | GPT-4o | 63.50 | 52.97 | 53.62 | 68.69 | 77.24 | **94.95** | 86.34 | 71.04 |
| | Gemini-2.5-pro | **68.96** | 59.20 | 61.19 | 74.91 | 83.92 | 93.55 | 83.40 | 75.02 |
| | DCR$^2$-AD-72B(Ours) | 68.71 | 62.13 | 69.23 | 80.16 | 86.37 | 93.07 | 89.23 | 78.42 |
| 1-shot | GPT-4o | 68.63 | **65.80** | 55.62 | 73.21 | 83.41 | **94.98** | 82.80 | 74.92 |
| | Gemini-2.5-pro | 67.73 | 62.07 | 64.73 | 76.45 | 84.91 | 93.25 | 83.96 | 76.16 |
| | DCR$^2$-AD-72B(Ours) | **74.37** | 64.60 | 71.71 | 82.23 | 87.17 | 93.24 | **89.28** | 80.37 |
| 1-shot with domain knowledge | GPT-4o | 69.36 | 65.41 | 55.26 | 78.78 | 78.92 | 95.40 | 86.59 | 75.68 |
| | Gemini-2.5-pro | **74.81** | 73.55 | 67.52 | 82.39 | 87.49 | 96.33 | 85.54 | 81.09 |
| | DCR$^2$-AD-72B(Ours) | 73.77 | **75.65** | **73.72** | **85.27** | **88.47** | **96.85** | **89.76** | **83.36** |

