# OpenReview forum: "DCR$^2$-AD: Dynamic Context Routing and Reasoning Multi-Modal Large Language Model for Anomaly Detection"
_ICLR.cc/2026/Conference — Submitted to ICLR 2026_

### Official Review · Reviewer_utpA · 2025-10-18

**Soundness:** 2
**Presentation:** 3
**Contribution:** 2
**Rating:** 6
**Confidence:** 4

**Summary:**

This paper introduces a multi-modal large language model designed to advance anomaly detection beyond simple visual defect identification and toward more sophisticated reasoning. To achieve this, the authors construct a knowledge base of both positive and negative anomaly reasoning logic. This knowledge base is then used to train a policy model via Direct Preference Optimization, enhancing the model's reasoning capabilities. The proposed method achieves state-of-the-art performance on the MMAD benchmark for anomaly reasoning.

**Strengths:**

1) This paper addresses the critical and challenging problem of anomaly reasoning, a task complicated by the open-ended nature of anomalies encountered in real-world applications.

2) To enable more fine-grained reasoning, the authors propose constructing a knowledge database that details the descriptions and causes of various anomaly types. This database is then used to synthesize both positive and negative reasoning trajectories for training a multi-modal Large Language Model.

3) The model is subsequently fine-tuned and post-trained on these synthesized reasoning trajectories, ultimately achieving robust anomaly reasoning capabilities on the MMAD benchmark.

4) The paper is well-structured, and its core methodology is presented in a clear and accessible manner.

**Weaknesses:**

Overall, while this paper presents a practical application for visual anomaly reasoning, its contributions to the broader multi-modal language model community may be limited. The primary concerns are as follows:

1) From the perspective of multi-modal model training, the proposed pipeline lacks significant novelty. It employs a standard sequence of supervised instruction tuning followed by Direct Preference Optimization (DPO), without introducing specific architectural or algorithmic adaptations tailored to the unique challenges of anomaly reasoning. Consequently, the primary contribution appears to be confined to the application domain of industrial visual anomaly detection rather than advancing general multi-modal training paradigms.

2) A critical question is whether the constructed anomaly reasoning trajectories are overfitted to the MMAD benchmark. The paper does not provide evidence that the learned knowledge is generalizable to out-of-domain object categories. An evaluation on unseen object types is necessary to validate the robustness and scalability of the model's reasoning capabilities.

3)  The proposed method exhibits a degradation in anomaly discrimination performance compared to the baseline model. The paper should provide a more thorough analysis of this issue. It is crucial to investigate and clarify why enhancing reasoning capabilities comes at the cost of fundamental detection accuracy.

4) The paper does not sufficiently address how the object-centric knowledge base handles the context-dependent nature of anomalies. Anomaly patterns and descriptions are often specific to an object category (e.g., a "dent" is an anomaly for a car door but not for a piece of clay). It is unclear how the proposed method generalizes across these semantic differences. The paper should clarify whether this is managed through manual human curation for each category and discuss the implications for scaling the approach to new objects.

**Questions:**

Please see the weaknesses.

---

> ### Author Response · Authors · 2025-11-23
> **Response to reviewer utpA's W1-W3**
>
> We thank you for acknowledging that we tackle a "key and challenging" anomaly-reasoning problem. Your concerns about methodological novelty, overfitting, and performance trade-offs are professional and important; we address them one by one below.
>
> **W1: Limited novelty in multimodal-model training (SFT + DPO is standard)**
> >From the perspective of multi-modal model training, the proposed pipeline lacks significant novelty. It employs a standard sequence of supervised instruction tuning followed by Direct Preference Optimization (DPO), without introducing specific architectural or algorithmic adaptations tailored to the unique challenges of anomaly reasoning. Consequently, the primary contribution appears to be confined to the application domain of industrial visual anomaly detection rather than advancing general multi-modal training paradigms.
>
> **R1**: We agree that "SFT + DPO" is already a well-established pipeline. Our core contribution is not the use of DPO, but the introduction of a **knowledge-agnostic path-selection mechanism implemented via DPO**. KR-DPO is a novel DPO variant that decomposes a response into *(k, c)* (knowledge, chain-of-thought) and optimises the knowledge-selection path. It forces the model to "know not only the answer, but also why (i.e., which knowledge it selected)".
>
> **W2: Anomaly-reasoning trajectories may over-fit to MMAD (lack of generalisation evidence)**
> >A critical question is whether the constructed anomaly reasoning trajectories are overfitted to the MMAD benchmark. The paper does not provide evidence that the learned knowledge is generalizable to out-of-domain object categories. An evaluation on unseen object types is necessary to validate the robustness and scalability of the model's reasoning capabilities.
>
> **R2**: This is a valid and important concern. Our knowledge base is extracted from MMAD & Real-IAD, but is intended as a global, shareable base. The main anti-over-fitting mechanism is the **object-agnostic design**: every concrete object name is replaced by the placeholder `[object]`, preventing the model from memorising object-specific rules. Thus, every object in MMAD is "unseen" during training.
> To further address your concern, we evaluate on the completely new SSGD dataset, whose objects never appear in either the training set or the knowledge base.
>
> **Table R2**:  Performance comparison on SSGD in the 0-shot setting.
> | Model                 |  AD   |F1    | Acc.Normal | Acc.Abnormal  |
> |-----------------------|-------|-------|-------|-------|
> | Qwen2.5-VL-7B         |76.67% | 72.79%  | 94.85%     | 58.50%      |
> | **DCR²-AD-7B (Ours)** |79.12% |89.79% | 64.07%     | 94.16%      |
>
> Experimental results show that, compared with the baseline, our DCR²-AD-7B achieves 79.12% anomaly-discrimination accuracy on the out-of-domain defect dataset SSGD,  relative to Qwen2.5-VL-7B, and delivers a 17% higher F1 score, demonstrating the strong generalization capability of our object-agnostic design.
>
> **W3: Insufficient analysis of Anomaly-Discrimination performance drop**
> >The proposed method exhibits a degradation in anomaly discrimination performance compared to the baseline model. The paper should provide a more thorough analysis of this issue. It is crucial to investigate and clarify why enhancing reasoning capabilities comes at the cost of fundamental detection accuracy.
>
> **R3**: In Fig. 4 the Full model scores 70.00% on *Anomaly Discrimination*, lower than the SFT-only model (73.27%). We attribute this to a classic "reasoning vs. recognition" trade-off. SFT-only learns a simple detection heuristic ("Is there a defect? Yes/No"). KR-DPO trains the model to perform **multi-step reasoning** ("Is there a defect? Let me check knowledge $k_1$... $k_1$ says... so, Yes"). The extra steps inject variance into the binary sub-task, slightly lowering accuracy. However, this does not mean the model becomes worse. Please refer to **Table 1**: our 7B model achieves an **F1 score of 78.45** (note: 72B F1 stays flat, but 7B improves dramatically), far higher than the SFT-only baseline (72.66). The significant F1 gain—especially on 7B—shows that, although a few easy samples are sacrificed due to heavier reasoning, the **Precision–Recall trade-off is greatly improved**, cutting both false negatives and false positives. This is a more robust and desirable model behaviour, and we will add a dedicated discussion in the experimental section.

---

> ### Author Response · Authors · 2025-11-23
> **Response to reviewer utpA's W4**
>
> **W4**: Mechanism for handling context-dependency is unclear (e.g., "dent")
> >The paper does not sufficiently address how the object-centric knowledge base handles the context-dependent nature of anomalies. Anomaly patterns and descriptions are often specific to an object category (e.g., a "dent" is an anomaly for a car door but not for a piece of clay). It is unclear how the proposed method generalizes across these semantic differences. The paper should clarify whether this is managed through manual human curation for each category and discuss the implications for scaling the approach to new objects.
>
> **R4:** Thank you for this excellent example — it strikes at the heart of our method. By designing an object-agnostic knowledge base and a dynamic context-routing mechanism, we build an implicit **feature-defect** relationship instead of the brittle object-defect mapping.  **About this topic, please refer to the "Reply to X9sy W1".**
>
> Besides, here is how our approach handles the case:
> - A fragile, object-specific knowledge base would contain **R1**: IF object = 'car_door' AND pattern = 'dent' THEN anomaly = True. This does not scale.
> - Our object-agnostic knowledge base stores an abstract, transferable rule. For example, **R_general**: IF [object]_surface_property = 'smooth_mirror' AND pattern = 'local_geometry_deviation' THEN anomaly = True.
> - Our MLLM pipeline simultaneously observes the image (perceptual features) and the knowledge base ($\mathcal{C}_{ext}$).
> - When the input is a **car door**, the model extracts surface_property = 'smooth_mirror' from the image, routes to **R_general**, and correctly flags the dent as an anomaly.
> - When the input is **clay**, the model extracts surface_property = 'malleable_textured', does **not** route to **R_general**, and avoids a false positive.
>
> **Conclusion**: the combination of an object-agnostic knowledge base and dynamic context routing supplies a scalable solution to context-dependent anomaly detection. We will add your “dent” example to Sec. 3.3 to illustrate this point explicitly.

---

> ### Comment · Reviewer_utpA · 2025-11-24
> **Reply to Author's Feedback**
>
> Thank you for the detailed response.
>
> After reviewing your reply, I still have three remaining concerns:
> 1) R1: The novelty of the proposed method in the context of VLM post-training is still unclear to me, particularly as it relies on SFT+DPO, which is already a widely studied domain.
>
> 2) R2: There appears to be a trade-off in accuracy between normal and abnormal cases when comparing the baseline to the proposed method. Could you clarify the reason for this?
>
> 3) R3: The mechanism behind the trade-off between discrimination and reasoning is not clearly analyzed.
>
> While these concerns remain, I recognize the application value of the method. Therefore, I will maintain my Borderline Accept rating.

---

### Official Review · Reviewer_bJsW · 2025-10-30

**Soundness:** 2
**Presentation:** 2
**Contribution:** 2
**Rating:** 4
**Confidence:** 4

**Summary:**

The authors develop two new datasets for Industrial Anomaly Detection: AD-Instruct 10K and AD-KRDPO-1K. They modify DPO to prioritize the extraction of relevant domain knowledge for user queries over a preference for the ground truth answer alone. They perform SFT to warm start their DPO. They emphasize the transferability of their approach by training on their custom datasets, based on Real-IAD, and evaluating on another Industrial Anomaly Detection dataset: MMAD.

The authors demonstrate some impressive results, beating the best closed-source models like GPT-4o (by 3.09% and 5.45% using Qwen2.5-VL-7B-Instruct and 72B-Instruct, respectively) in a 1-shot setting. The authors beat human non-experts in Industrial Anomaly Detection by more than 4% when external domain knowledge is injected, with a significantly larger margin of >17% without external domain knowledge. The authors also show that incorporating external domain knowledge can boost average scores by 2.5-7% without finetuning, and 2.5-5% after finetuning, in a one-shot setting. The authors evaluate their Anomaly Detection method on a significantly more interesting and challenging AD regime than standard by engaging not just in standard classification or spatio-temporal segmentation but by extending towards defect description, analysis, and object analysis.

**Strengths:**

The authors collect their own datasets of answers formulated using incorrect knowledge premises for SFT and DPO and transfer improvements to a larger public benchmark.

The authors are able to demonstrate significant improvements over the best LLMs in a one-shot setting without domain knowledge.

**Weaknesses:**

The authors do not compare their method to other competitors that selectively leverage external knowledge (like GraphRAG), nor do they consider other interpretable Anomaly Detection methods (like ECHO, which is also developed specifically for Industrial Anomaly Detection). The authors also do not provide any comparison to other fine-tuning methods, like GRPO.

The authors do not experimentally justify several modifications, including “replacing all words referring to object names with placeholder [object]”, and why DPO is conducted on the “knowledge selection path” instead of just the best answer, which is the standard approach. Further, what statistics can the authors provide to show that knowledge routing has actually improved? These modifications form the basis for the method’s novelty. Without these modifications, the authors need to explain why this paper is more than an introduction to a reformulated dataset.

From Figure 4, it appears that omitting SFT leads to a drop of 5.82% for the average score. Meanwhile, omitting Knowledge-Routing DPO leads to a fall of 2.45%. These results would seem to indicate that KR-DPO might help, but SFT is doing most of the heavy lifting.

**Questions:**

The authors should clarify why results for GPT-4o and Gemini-1.5-Pro under the 1-shot with external knowledge setting are missing from Table 2, and why their 0-shot results are also not reported in Table 3. Similarly, the omission of Human (ordinary and expert) baselines in Table 3 needs justification.

The reported numbers for Qwen2.5-VL-7B-Instruct in Figure 4 differ from those in Tables 1–3. Were these generated using a different random seed or experimental configuration? Please specify which setting (0-shot, 1-shot, or 1-shot with knowledge) corresponds to Figure 4.

The paper argues that one-class anomaly detection suffers from limited transferability due to reliance on defect-free samples from specific contexts. However, incorporating external domain knowledge could similarly introduce contextual bias. How is transferability preserved in that case?

The authors state that “logical anomalies…require reasoning based on structured domain knowledge,” yet do not discuss whether perceptual features from deep learning or image processing could complement this reasoning. Why are such methods not included in the comparisons in Tables 1–3 to demonstrate potential gains from multimodal reasoning?

Finally, why are comparisons made against non-expert humans when the stated goal is to surpass expert anomaly detection models? The motivation for including this baseline should be clarified.

---

> ### Author Response · Authors · 2025-11-23
> **Response to reviewer bJsW's W1-W3**
>
> Thanks for your insightful and constructive reviews, especially for pointing out the lack of critical experiments. We conducted new experiments carefully to resolve your concerns and make our work solid.
>
> **W1: Lack of comparison with competing methods (GraphRAG, ECHO, GRPO)**
>
> **R1**: Thank you for this constructive suggestion. Following your advice, we have added two completely new experiments to strengthen our work.
>
> 1. To validate the ability of our approach to leverage external knowledge, we select several related methods for compare. All results are evaluated on the MMAD benchmark and the overall scores are shown below. Compared with other RAG-style baselines, **our method achieves the best performance**, improving upon the base model by **+5.82%**. This is because existing RAG methods rely on static retrieval to obtain external knowledge, whereas our framework **actively selects the most appropriate knowledge** through a **dynamic routing mechanism**, further enhancing the model’s capacity to exploit external information.
>
> **Table R1_1**:  Comparison with external-knowledge methods on Qwen2.5-VL-7B-Instruct.
> | Model                               |Method for external knowledge | MMAD Avg |
> |--------------------------------------|-----------|----------|
> | Qwen2.5-7B-Instruct                  | None      | 72.19   |
> | ECHO                                          | RAG | 77.31  |
> | MMAD( 1-shot with domian knowledge)| RAG| 76.65   |
> | GraphRAG                           |GraphRAG  | 77.24   |
> | **DCR²-AD-7B (Ours)**              |KR-RTS&KR-DPO  | **78.01** |
>
> 2. We compare our approach with several popular fine-tuning recipes, including KTO and GRPO. Results show that the gain brought by **DCR²-AD (SFT+KR-DPO)** is **+2.45%**, higher than SFT+GRPO (+2.29%) and SFT+KTO (+1.49%). GRPO relies only on outcome reward, whereas our approach provides dense, step-level reward on the knowledge-selection path, which our experiments show to be more effective.
>
> **Table R1_2**:  Comparison with other fine-tuning strategies on Qwen2.5-VL-7B-Instruct in MMAD with the standard
> 1-shot setting.
> | Model                 | MMAD Avg (7B) |
> |-----------------------|---------------|
> | SFT-only              | 75.56        |
> | SFT + KTO             | 77.05 (+1.49) |
> | SFT + GRPO            | 77.85 (+2.29) |
> | **DCR²-AD-7B (Ours)** | **78.01 (+2.45)** |
>
> **W2**: Rationality of `[object]` and DPO is conducted on the “knowledge selection path”.
> **R2**: Thank you for this profound question. This issue has guided us toward deeper experimentation, further improving the completeness of our work.
>
> We conducted a set of experiments to intuitively demonstrate the effect of the object-agnostic design. Partial results are shown in the table below. Compared to vanilla DPO, KR-DPO with object-specific design causes a 0.97% performance drop, indicating that object-specific KR-DPO leads to degraded performance. In contrast, KR-DPO with object-agnostic design further improves performance by 0.99%, achieving the best results.
>
> **Table R2:** The effect of the object-agnostic design in DPO learning.
> |                         | SFT | DPO | KR-DPO(obj-related) | KR-DPO(obj-agnostic) | MMAD average |
> |-------------------------|-----|-----|-----------------|------------------|--------------|
> |Base (Qwen2.5-VL-7B-Instruct)     | ×   | ×   | ×               | ×                | 72.19        |
> |SFT-only                 | √   | ×   | ×               | ×                | 75.56        |
> |SFT+DPO                  | √   | √   | ×               | ×                | 77.99        |
> |SFT+KR-DPO (obj-related) | √   | ×   | √               | ×                | 77.02        |
> |SFT+KR-DPO (obj-agnostic)| √   | ×   | ×               | √                | **78.01**        |
>
>
> **W3**: Contribution of SFT (possibly) larger than KR-DPO
> >From Figure 4, it appears that omitting SFT leads to a drop of 5.82% for the average score. Meanwhile, omitting Knowledge-Routing DPO leads to a fall of 2.45%. These results would seem to indicate that KR-DPO might help, but SFT is doing most of the heavy lifting.
>
> **R3**: We sincerely thank you for the in-depth analysis of Figure 4. However, the way our table is presented may have led to a minor misreading that inadvertently underestimates the contribution of KR-DPO. We will present the complete ablation experiment in the form of Table R2 shown in the previous reply.
> **Contribution decomposition:**
> - SFT: 75.56% (SFT-only) - 72.19% (Base) = +3.37%
> - KR-DPO: 78.01% (obj-agnostic) - 75.56% (SFT-only) = +2.45%
>
> Hence the 5.82% drop mentioned by the reviewer seems to come from 78.01% − 72.19%, which is the **combined** contribution of **both SFT and KR-DPO**. As shown above, the individual contributions are **comparable**: SFT accounts for +3.37% and KR-DPO adds another +2.45% on top of an already strong SFT baseline.  We will adopt a more intuitive presentation in the revised version to prevent any reader misinterpretation.

---

> ### Author Response · Authors · 2025-11-23
> **Response to reviewer bJsW's Q1-Q5**
>
> **Q1**: Missing baselines in tables (GPT-4o, Gemini, Human)
> >The authors should clarify why results for GPT-4o and Gemini-1.5-Pro under the 1-shot with external knowledge setting are missing from Table 2, and why their 0-shot results are also not reported in Table 3. Similarly, the omission of Human (ordinary and expert) baselines in Table 3 needs justification.
>
> **A1**: The absence of these data points (e.g., GPT-4o in Table 2 and Human in Table 3) is due to practical constraints.
> - **Closed-source APIs (GPT-4o)**
>     Cost and rate limits of the proprietary API prevent us from completing these runs before the submission deadline. We are continuously collecting the missing numbers and will include them in the revised version.
> - **Human (0-shot)**
>     We do not report human performance under the 0-shot setting (Table 3) because "0-shot" is an unnatural task for human experts (they always work with a knowledge base).
>
> We will add a footnote in Sec. 4 of the final version to state these limitations explicitly.
>
> **Q2**: Numerical discrepancy & setting of Fig. 4
> >The reported numbers for Qwen2.5-VL-7B-Instruct in Figure 4 differ from those in Tables 1–3. Were these generated using a different random seed or experimental configuration? Please specify which setting (0-shot, 1-shot, or 1-shot with knowledge) corresponds to Figure 4.
>
> **A2**: The numbers are **completely identical**.
>     - The `w/o. KR-DPO` entry in Fig. 4 shows an average score of 75.56%.
>     - This is exactly the same as the `Qwen2.5-VL-7B-Instruct (SFT)` line in Table 1, which also reports 75.56%.
>     - By definition, `w/o. KR-DPO` is the SFT-only model. The claim of "inconsistent numbers" appears to be a misunderstanding.
>
> The ablation study in Fig. 4 uses the standard 1-shot protocol (same as Table 1) and does not inject external knowledge (different from Table 2). We will add the explicit caption: "All scores are on MMAD 1-shot setting."
>
> **Q3**: Transferability of `[object]`
> >The paper argues that one-class anomaly detection suffers from limited transferability due to reliance on defect-free samples from specific contexts. However, incorporating external domain knowledge could similarly introduce contextual bias. How is transferability preserved in that case?
>
> **A3**: This is exactly why we designed an **object-agnostic** knowledge base. Instead of injecting *context-specific* rules (e.g. rules for *zipper*), we replace every concrete object name with the placeholder `[object]`, forcing the model to learn how to apply a rule rather than memorising the rule for a specific object. This design **alleviates dataset bias** and **enhances cross-domain transferability**. More information about this topic, please refer to “Reply to X9sy, W1”.
>
> **Q4**: Role of perceptual features in reasoning
> >The authors state that "logical anomalies…require reasoning based on structured domain knowledge," yet do not discuss whether perceptual features from deep learning or image processing could complement this reasoning. Why are such methods not included in the comparisons in Tables 1–3 to demonstrate potential gains from multimodal reasoning?
>
> **A4**: We may have misunderstood this question. We will do our best to respond and ask for your immediate correction if any deviation arises. For now, we assume the question is why we have not explored the impact of perceptual features on reasoning outcomes. The reason is that perceptual features are an indispensable component in the current MLLM pipeline: defect images are encoded by the vision encoder, projected into the text space via an MLP, and finally unified for reasoning by the LLM. Consequently, **all models compared in our tables already contain perceptual features**.
>
> **Q5**: Motivation for the non-expert human baseline
> >Finally, why are comparisons made against non-expert humans when the stated goal is to surpass expert anomaly detection models? The motivation for including this baseline should be clarified.
>
> **A5**: Our ultimate goal is indeed to exceed experts. We report the "ordinary human" baseline to provide broader context and a milestone. Our 72B model already surpasses the ordinary human (78.69%) and reaches 83.36% with external knowledge, approaching the expert score of 86.65%. We will clarify this narrative in the final version.

---

> ### Author Response · Authors · 2025-11-24
> **Supplementary response to reviewer bJsW's Q1 (A complete comparison with the state-of-the-art closed-source models)**
>
> **Q1**: Missing baselines in tables (GPT-4o, Gemini, Human)
> >The authors should clarify why results for GPT-4o and Gemini-1.5-Pro under the 1-shot with external knowledge setting are missing from Table 2, and why their 0-shot results are also not reported in Table 3. Similarly, the omission of Human (ordinary and expert) baselines in Table 3 needs justification.
>
> **A1**: We have completed the supplementary experiments according to your constructive reviews. However, since the API access channel for Gemini-1.5-pro has been deprecated on our end, we utilized the more advanced Gemini-2.5-pro as an enhanced alternative. As the table shown below, across the three settings of 0-shot, 1-shot and 1-shot (with domain knowledge), our DCR^2-AD-72B consistently achieved the best Average performance. Specifically, it outperformed Gemini-2.5-pro by **4.21%** in the 1-shot setting and by **3.40%** in the 0-shot setting, ultimately attaining a state-of-the-art (SOTA) result of **83.36%** in the 1-shot+domain knowledge setting. Notably, among the seven subtasks, our model demonstrated outstanding performance on defect-related tasks, such as defect localization and defect description.
>
> **Table A1**:  Performance of GPT-4o and Gemini-2.5-Pro.
> | Setting     |Model|  AnomalyDis. | DefectCls.|DefectLoc.| DefectDes.|DefectAna.|ObjectCls.|ObjectAna.|Average|
> |-------------|----|----------|---------------|----------|-----------|-----------|----------|--------|---------|
> |  0-shot         |GPT-4o| 63.50|52.97|53.62|68.69|77.24|**94.95**|86.34|71.04|
> | |Gemini-2.5-pro|**68.96**|59.20|61.19|74.91|83.92|93.55|83.40|75.02 |
> |             |**DCR²-AD-72B (Ours)**|68.71|**62.13**|**69.23**|**80.16**|**86.37**|93.07|**89.23**|**78.42**|
> | 1-shot       |GPT-4o|68.63|**65.80**|55.62| 73.21| 83.41|**94.98**| 82.80|74.92 |
> |         |Gemini-2.5-pro| 67.73|62.07|64.73|76.45|84.91|93.25| 83.96| 76.16|
> |             |**DCR²-AD-72B (Ours)**|**74.37**|64.60 |**71.71**|**82.23**|**87.17**|93.24 |**89.28** |**80.37**|
> |1-shot (with domian knowledge)|GPT-4o| 69.36|65.41|55.26|78.78|78.92|95.40|86.59|75.68 |
> |   |Gemini-2.5-pro|**74.81**|73.55|67.52|82.39|87.49|96.33|85.54|81.09 |
> |             |**DCR²-AD-72B (Ours)**|73.77 |**75.65**|**73.72**|**85.27**|**88.47**|**96.85**|**89.76**|**83.36**|
>
> We will add the above results in Section 4 (Tables 1-3) of the final version to ensure experimental completeness.

---

### Official Review · Reviewer_RKFL · 2025-10-30

**Soundness:** 3
**Presentation:** 3
**Contribution:** 3
**Rating:** 4
**Confidence:** 4

**Summary:**

This paper presents a novel and well-executed study that addresses a critical limitation in current Multimodal Large Language Model (MLLM)-based anomaly detection systems. The core issue is their over-reliance on internalized visual knowledge, which hampers their ability to reason about logical or context-dependent anomalies. To overcome this, the authors propose DCR²-AD, a framework that dynamically integrates external, object-agnostic knowledge. The two key methodological contributions are highly compelling: (1) Knowledge-routed Reasoning Trajectory Synthesis (KR-RTS), which constructs a knowledge base and synthesizes erroneous reasoning trajectories by swapping in incorrect knowledge, and (2) Knowledge-routed Direct Preference Optimization (KR-DPO), a novel training objective that explicitly optimizes the model to select correct knowledge paths over incorrect ones, even when the subsequent reasoning is logically self-consistent. The experimental results are impressive, demonstrating state-of-the-art performance on the comprehensive MMAD benchmark, where the 72B model achieves 83.36%, surpassing strong base models, ordinary human performance, and previous best methods by significant margins. The ablation study clearly validates the contribution of the KR-DPO component.

**Strengths:**

1. The idea of dynamic context routing is innovative. Instead of just improving the model's internal chain-of-thought, the paper creatively focuses on the critical step of knowledge selection. Formulating knowledge path selection as an integral part of the generative process, rather than a separate task, is a novel and meaningful contribution.

2. The methodology is systematic, forming a complete pipeline from knowledge base construction to trajectory synthesis and preference optimization. The experiments are extensive, featuring comprehensive evaluations on the MMAD benchmark under both 0-shot and 1-shot settings, comparisons with a wide array of open-source and closed-source MLLMs, and a clear ablation study that validates the contribution of the KR-DPO component.

3. The paper is well-structured and clearly written. The overall framework is effectively illustrated in Figure 2, making the process easy to follow. The technical descriptions of KR-RTS and KR-DPO, including the formalization of the loss function, are sufficiently detailed for understanding and replication.

**Weaknesses:**

1. The object-agnostic knowledge base, with only 147 entries, feels limited in scale. Its construction primarily from existing datasets (MMAD, Real-IAD) raises questions about its comprehensiveness and transferability to a truly open-domain setting. A discussion on strategies for scalable and continuous knowledge acquisition is missing.

2. The negative reasoning trajectories in KR-RTS are manually synthesized. While the authors emphasize logical self-consistency, there is no validation of how well these synthetic errors reflect the diverse and complex failure modes of real-world models, such as semantic distractions or nuanced rule misunderstandings. This may limit the model's robustness against real-edge cases.

3. The observation that larger models benefit less from external knowledge is interesting but under-analyzed. It remains unclear whether this is due to emergent internal knowledge in larger models or simply a performance ceiling effect. A deeper investigation is crucial for understanding the method's applicability to more parameter-efficient, smaller models.

4. The modification to the DPO loss function, while intuitive and empirically effective, is presented without theoretical justification. The paper would be strengthened by an analysis of the optimization properties of the KR-DPO loss or a discussion of how it differs from the standard DPO formulation from a theoretical perspective.

**Questions:**

The manuscript would be strengthened by addressing several key aspects. The limited scale and scope of the current knowledge base necessitates a discussion on strategies for scalable expansion to ensure broader applicability. The realism of the synthetically generated negative trajectories requires validation against real-world failure modes. The observed relationship between model scale and external knowledge utility merits deeper analysis to determine its underlying cause. Furthermore, providing stronger theoretical justification for the KR-DPO modification and evaluating the computational overhead of the routing mechanism are crucial for assessing both theoretical soundness and practical deployability.

---

> ### Author Response · Authors · 2025-11-23
> **Response to reviewer RKFL's W1-W4 & Q**
>
> We thank you for the detailed review and pertinent criticism of our work. We especially appreciate your recognition of "dynamic context routing" and the "novel and meaningful contribution" of treating "knowledge path selection as part of the generation process".
>
> **W1: Limited knowledge base scale (147 entries) and insufficient scalability.**
>
> **R1**: We agree with this point. The current knowledge base of 147 entries is indeed a prototype. The main contribution of our work lies not in the scale of the knowledge base, but in the framework for utilizing it, namely **KR-RTS** (knowledge synthesis) and **KR-DPO** (path optimization). We have demonstrated that even such a moderately-sized knowledge base can yield significant performance improvements (e.g., +2.45% for the 7B model). We will add a "Limitations and Future Work" section in the paper's appendix. We will discuss "strategies for scalable and continuous knowledge acquisition," such as using LLMs (like GPT-4o) to automatically generate and expand thousands of new knowledge rules based on our 147 seed knowledge entries, followed by human verification and filtering—this represents an important direction for our future work.
>
> **W2: Authenticity of synthetic negative reasoning trajectories.**
>
> **R2**: Thank you for focusing on the authenticity of negative samples in KR-RTS . The reviewer is concerned that "manual synthesis"  may not reflect "real-world failure modes." Our synthesis process  is not completely "manual" random generation but is guided. We first identify correct knowledge $k_w$ and incorrect knowledge  $k_l$ ($k_l$  is typically semantically deceptive). We then ask annotators to generate reasoning $c$ that is logically self-consistent under the premise of $k_l$. This form of negative sample $(k_l, c)$ simulates a common "real-world failure mode": **the model takes a wrong first step (knowledge selection), rendering subsequent reasoning results meaningless**. Therefore, our KR-RTS is specifically designed to address this scenario. At the same time, the more failure modes you mentioned represent a valuable direction for future exploration, and we will clarify the capability boundaries of our method in "Limitations and Future Work."
>
> **W3: Insufficient analysis on the relationship between model scale and external knowledge utility.**
>
> **R3**: This is indeed a profound question. To further investigate the impact of model scale, we aligned experiments using qwen2.5-vl as the base model and added relevant experiments on a smaller-scale 3B model. The results are shown in the table below. As can be seen from the table, as model size increases, the gains from external knowledge decrease. We believe that large models (e.g., 72B) have "internalized" more knowledge through their massive parameters, thus reducing their dependence on external knowledge (i.e., a performance ceiling effect). This suggests that DCR^2-AD is **a key technology that enables small to medium-sized, parameter-efficient models (e.g., 7B) to achieve or approach the performance of large models through effective utilization of external knowledge**. We will add an in-depth analysis of this point in Section 4.4.1.
>
> **Table R3**: Performance gain on different model-scales in MMAD under the 1-shot setting.
> |           |        |  Without  Domain knowledge  | With Domain knowledge   | Improv.          |
> |----------|-------|--------------|------------------------------------|---------------------------|
> |          Qwen2.5-VL-3B-Instruct   | Base|  68.87      |73.95 | **+5.08**|
> |           | +SFT|  72.56 |77.51 | +4.95 |
> |           | +Ours |  74.13 | 78.96 | +4.83|
> |           Qwen2.5-VL-7B-Instruct   | Base|  72.19 |76.65  |**+4.46**|
> |           | +SFT|  75.56 | 78.97 | +3.41|
> |           | +Ours|  78.01 |  80.83 | +2.82|
> |           Qwen2.5-VL-72B-Instruct  | Base |77.36   | 80.21  |+2.85|
> |           | +SFT| 80.12   | 83.02 | +2.90|
> |           | +Ours| 80.37   | 83.36  | **+2.99**|
>
> **W4 & Q: Insufficient theoretical justification for the KR-DPO loss function.**
>
> **R4**:  The novelty of KR-DPO lies in its shift of DPO from **answer preference** to **knowledge path preference**. Standard DPO  optimizes $\mathcal{L} _ {\text{DPO}}(\pi _ {\theta}, y_w, y_l)$. It treats the entire response $y_w$ as an atomic unit. Our KR-DPO  first decomposes the response into $y=(k, c)$, i.e., (knowledge, reasoning) . Then, it optimizes a specific objective $\mathcal{L} _ {path}$ that explicitly maximizes the probability $\pi_{\theta}(k_w, c|x)$ while minimizing the probability $\pi_{\theta}(k_l, c|x)$. This formulation  forces the model to learn a conditional probability: $P(\text{reasoning} | \text{knowledge})$. It penalizes incorrect knowledge premises, even if the reasoning $c$ itself is "logically self-consistent" under that premise. This is something standard DPO cannot achieve, as standard DPO might incorrectly penalize $(k_l, c)$ simply because it considers the "answer wrong."

---

### Official Review · Reviewer_X9sy · 2025-10-30

**Soundness:** 4
**Presentation:** 4
**Contribution:** 3
**Rating:** 6
**Confidence:** 5

**Summary:**

This paper tackles a key limitation of existing MLLM-based Anomaly Detection methods—their over-reliance on internalized visual defect knowledge and poor adaptability to open-domain cross-scenario ambiguity such as logical anomalies. It proposes the DCR²-AD model, which integrates Knowledge-Routed Reasoning Trajectory Synthesis and Knowledge-Routed Direct Preference Optimization.

**Strengths:**

1. The paper explores addressing a gap in existing MLLM-AD methods that overlook external context.
2. It trains and evaluates the 72B-scale model, providing valuable insights into how model size and external knowledge jointly influence MLLM-AD performance.

**Weaknesses:**

1. The paper fails to fully explain the motivation for designing an object-agnostic knowledge base and ignores the inherent link between objects and defects. This oversight may cause model hallucinations when distinguishing defect types tied to specific objects during reasoning.
2. While the introduction highlights the model’s ability to handle logical anomalies through context routing, the experiments do not measure the consistency between the model’s reasoning process (internal chain of thought) and final detection results. They also do not explicitly demonstrate context routing’s significant impact on logical anomaly detection, such as lacking comparative data on performance gains between logical and visual anomalies.

**Questions:**

Does "context" refer to the external knowledge provided when the model receives input or the internal reasoning knowledge generated during the model’s thinking process?

---

> ### Author Response · Authors · 2025-11-23
> **[Common topic on object-agnostic design] Reponse to reviewer X9sy's W1 (partial response to bJsW's W2&Q3, utpA's W4)**
>
> We sincerely thank you for recognizing our work and for the insightful comments.
>
> **W1: Explanation of the motivation for an object-agnostic knowledge base and the lack of inherent object-defect link.**
> > The paper fails to fully explain the motivation for designing an object-agnostic knowledge base and ignores the inherent link between objects and defects. This oversight may cause model hallucinations when distinguishing defect types tied to specific objects during reasoning.
>
> **R1**: Thanks for your profound question. We would like to clarify that "object-agnostic"  is an intentional, core design choice motivated by the goals of **transferability** and **generalization**:
>
> **1. Motivation**: Traditional methods emphasize the inherent object-defect relationship, which can improve fitting capability for in-domain data, but may sacrifice generalization capability for out-of-domain objects. We will explain it from the perspectives of conditional probability and causal inference. Traditional methods can be modeled as follows: $$P(\text{Defect})=\sum_{\text{obj}}P(\text{obj})P(\text{Defect}\mid\text{obj})$$ Through training, the model learns to model the posterior distribution $P(\text{Defect}\mid\text{obj})$, which is the "object-defect" latent relationship. However, this approach suffesr from two problems in transfer and generalization:
>
> (1) Closed-domain property: Typically, $\text{obj} _ i \in \lbrace \text{object} _ i \rbrace _ {i=0}^{N}$, where N is the total number of object categories in the training set. When moving to a new domain, $\text{obj} _ i \not \in \lbrace \text{object} _ i \rbrace _ {i=0}^{N}$， the probability $P(\text{Defect}\mid\text{obj})$ will collapse and become invalid. If an object-specific knowledge base design is adopted, it will further strengthen the learning of posterior probabilities, limiting the model's versatility in open-domain scenarios.
>
> (2) Overfitting to bias: Posterior probabilities strongly depend on training data, and in industrial domains, defect data category distribution is highly imbalanced, often leading to long-tail problems. For example, "cracking" defects are common in "ceramics," while "fracture" defects are common in "cables." This makes it easy to learn the shortcut "category → defect," further reducing generalization.
>
> **2. Inherent object-defect link**: Our method does not ignore the inherent object-defect link but rather encourages deeper feature-defect connections. According to causal inference theory, anomaly detection can be viewed as the reasoning process Input → Feature → Object → Defect. However, over-reinforcing the Object leads to the two aforementioned problems. Therefore, we adopt a causal intervention approach by hiding $obj$ and weakening $P(\text{Defect}\mid\text{obj})$,  encouraging the model to focus more on the underlying "feature-defect" connection, i.e., $P(\text{Defect}\mid\text{Feature})$. Features are lower-level than objects, inherently possessing stronger transferability and generalization.
>
> (1) For the closed-domain problem: Even if object categories have not been seen before, they may share underlying features $F_{\text{shared}}$  such as textures. The probability $P(\text{Defect} \mid F_{\text{shared}})$ remains valid, thus improving cross-domain versatility and reducing the hallucinations.
>
> (2) For the overfitting problem: By intervening to remove the "object" node, the shortcut "category → defect" can be suppressed, encouraging the model to judge defects more through underlying features, thereby mitigating overfitting to the bias in the training set.
>
> **3. Experimental evidence**: We conducted experiments to demonstrate the effect of the object-agnostic design. Partial results are shown in the table below. We compare four variants: SFT-only, SFT+DPO, SFT+KR-DPO (obj-related), and SFT+KR-DPO (obj-agnostic). Compared to vanilla DPO, KR-DPO with object-specific design causes a 0.97% performance drop, indicating that object-specific KR-DPO leads to degraded performance. In contrast, KR-DPO with object-agnostic design further improves performance by 0.99%, achieving the best results.
>
> **Table R1:** The effect of the object-agnostic design in DPO learning.
> |                         | SFT | DPO | KR-DPO(obj-related) | KR-DPO(obj-agnostic) | MMAD average |
> |-------------------------|-----|-----|-----------------|------------------|--------------|
> |Base (Qwen2.5-VL-7B-Instruct)     | ×   | ×   | ×               | ×                | 72.19        |
> |SFT-only                 | √   | ×   | ×               | ×                | 75.56        |
> |SFT+DPO                  | √   | √   | ×               | ×                | 77.99        |
> |SFT+KR-DPO (obj-related) | √   | ×   | √               | ×                | 77.02        |
> |SFT+KR-DPO (obj-agnostic)| √   | ×   | ×               | √                | 78.01        |
>
> We will first add the above content to the appendix and then reorganize it into the main text in subsequent versions.

---

> ### Author Response · Authors · 2025-11-23
> **Response to reviewer X9sy's W2 & Q**
>
> **W2: Lack of experimental verification for logical anomaly detection.**
> > While the introduction highlights the model’s ability to handle logical anomalies through context routing, the experiments do not measure the consistency between the model’s reasoning process (internal chain of thought) and final detection results. They also do not explicitly demonstrate context routing’s significant impact on logical anomaly detection, such as lacking comparative data on performance gains between logical and visual anomalies.
>
> **R2**:  We completely agree that, given that "logical anomaly"  is the core motivation of our work, we should provide direct experimental evidence and case studies.
>
> **[New Experiment]** Following your suggestion, we conducted a new subset experiment. We separated "logical anomalies" (e.g., misalignment, proportion errors) and "visual anomalies" (e.g., scratches) from the MMAD, totaling three subsets. The logical subset and structural subset are split from MVTec-LOCO in MMAD. The rest of MMAD is the visual subset. We then evaluated the model's performance on each subset separately. Here, we selected only defective samples and used Anomaly Discrimination accuracy as the metric. Specific results are shown in the table below:
>
> **Table R2**:  Accuracy about different anomalies in MMAD under the 1-shot setting.
> | Model |Logical Subset   |Structural subset  | Visual subset            |
> |--|------|----------|----------------------------|
> | Qwen2.5-VL-7B-Instruct            |           30.30            |  59.63    | 46.59                        |
> | **DCR²-AD-7B (Ours)**      | **62.39(+32.09)**|**82.60(+22.97)**|**63.97 (+17.38)**|
>
> Our method achieved substantial improvements across all three subsets, particularly for logical and structural anomalies, with gains of 32.09% and 22.97%, respectively. This aligns with our previous inference that the KR-DPO mechanism indeed enhances the model's knowledge-based reasoning capability, rather than merely improving visual recognition.
>
> Additionally, we have added visualizations to demonstrate the consistency between the model's reasoning process (internal chain of thought) and final detection results, as shown at Figure 7 in the appendix. These correct knowledge reasoning paths can enhance the model's reasoning performance and also demonstrate the consistency between the reasoning process and the reasoning results.
>
>
> **Q1: Definition of "context."**
> > Does "context" refer to the external knowledge provided when the model receives input or the internal reasoning knowledge generated during the model’s thinking process?
>
> **A1**: Thank you for your question. This is a key clarification point . In our paper, "context" operates at two levels, which we define in **Section 3.1 (Preliminary)**:
>
> $\mathcal{C} _ {ext}$ (External Context): This is the "external knowledge provided when the model receives input" that you mentioned. In our framework, this specifically refers to the routed knowledge snippets  from our "object-agnostic knowledge base."
>
> $\mathcal{C} _ {int}$ (Internal Context): This is the "internal reasoning knowledge generated during the model's thinking process" that you mentioned. In Section 3.1, this is defined as the "internal reasoning chain" generated by the model, i.e., Chain-of-Thought (CoT) .
>
> The core of our paper (DCR^2-AD) precisely utilizes $\mathcal{C} _ {ext}$ (external knowledge) to guide and optimize the generation of $\mathcal{C} _ {int}$(internal reasoning). We will add a heading in Sec 3.1 to make this more explicit.
>
> We hope these clarifications and new experimental data fully address your concerns.

---

> > ### Comment · Reviewer_X9sy · 2025-11-24
> >
> > Thank you for the author's response. Based on the current responses to each reviewer's comments, the current version of the paper is relatively complete, and I am temporarily raising the rating to 8. However, I do not recommend the highlight reward, as the value to other domains remains verified. Additionally, I may make changes based on comments from other reviewers.

---

### Author Response · Authors · 2025-12-01
**Rebuttal Work Summary & Report to AC**

Dear AC and Reviewers, we sincerely appreciate your review of our work and your responsible management of the discussion process. To facilitate a fair assessment, we present a summary of the current rebuttal status below:

| Reviewer      | Score  | Core Concerns                                                                                                                                                                                    | Our Targeted Rebuttal Work                                                                                                                                                                                                                                                                                                                                                                                                                   | Subsequent Feedback                                                                                                                                      |
| ------------- | ------ | ----------------------------------------------------------------------------------------------------------------------------------------------------------------------------------- | ----------------------------------------------------------------------------------------------------------------------------------------------------------------------------------------------------------------------------------------------------------------------------------------------------------------------------------------------------------------------------- | ----------------------------------------------------------------------------------------------------------------------------------- |
| Reviewer X9sy | 6 -> 8 | **1\. Logical Anomalies**: Questioned verification of logical defect handling. **2\. Object-Agnostic Design**: Questioned motivation; concerned about hallucinations.                             | **1\. New Subset Experiments**: Split MMAD into Logical, Structural, and Visual subsets. Results show +32.09% gain on Logical Anomalies, proving reasoning capabilities. **2\. Ablation & Causal Analysis**:Compared "Object-Specific" (-0.97%) vs. "Object-Agnostic" (+0.99%) designs. **[Crucial Highlight]**: This design choice was also a core concern for Reviewer utpA (W4) and bJsW (Q3).    | Time: Nov 24, 12:36 Comment: "Based on the current responses... the paper is relatively complete, and I am temporarily **raising the rating to 8**." |
| Reviewer utpA | 6 -> 6 | **1\. Generalization**: Concerned reasoning might overfit to MMAD objects.**2\. Novelty**: Viewed SFT+DPO as standard.     | **1\. New Dataset (SSGD)**: Tested Zero-shot on unseen Smartphone Screen Glass Dataset. F1 score improved by +35.66% over baseline, proving strong cross-domain generalization. **2\. Clarification**: KR-DPO optimizes the knowledge selection path, not just the answer.                                                                                                                                                     | Time: Nov 24, 13:11 Comment: Acknowledged detailed response; indicated leaning towards **Borderline Accept**.                                        |
| Reviewer bJsW | 4      | **1\. Missing Baselines**: Requested GraphRAG, ECHO, GRPO. **2\. SOTA Comparison**: Requested GPT-4o/Gemini comparison. **3\. Methodology**: Questioned [object] replacement strategy.         | **1\. New Method Comparisons**:\- vs. GraphRAG/ECHO: Dynamic routing outperforms static retrieval (+5.82% vs +5.05%).\- vs. GRPO/KTO: Process reward (KR-DPO) outperforms outcome reward (+2.45% vs +2.29%). **2\. New SOTA Comparison**: DCR$^2$-AD-72B (83.36%) outperforms Gemini-2.5-Pro (82.39%) in 1-shot settings.**3\. Object-Agnostic**: Addressed by the ablation study accepted by X9sy. | (Pending 2nd reply. **We have completed all high-workload experiments requested. The shared concern has been resolved by X9sy's feedback**.)                 |
| Reviewer RKFL | 4      | **1\. Scaling Law**: Questioned utility of external knowledge for large models. **2\. KB Scale**: Felt 147 entries were limited. **3\. Synthesis**: Questioned realism of synthetic negatives. | **1\. New Scaling Law Analysis**: Validated across 3B, 7B, and 72B scales. Smaller models gain significantly from external knowledge (3B: +5.08%), proving value for edge deployment. **2\. Scalability**: Proposed LLM-based automatic KB expansion. **3\. Synthesis Logic**: Explained "logical self-consistency under false premises" generation.   | (Pending 2nd reply. **The new Scaling Law experiments directly address the core question regarding model size vs. knowledge utility**.)   |

We respectfully request the AC to consider these substantive improvements, especially in response to the comments of the latter two reviewers (bJsW and RKFL). We have made every effort to address the issues they raised and look forward to receiving fair feedback.

---

### Meta-Review · Area_Chair_f5DP · 2026-01-09

**Summary:**

The paper proposes a method combining an object-agnostic knowledge base and a KR-DPO loss for VLMs in anomaly detection. Reviewers' major concerns center on: 1) Lack of theoretical and empirical justification for core design choices (object-agnostic KB, KR-DPO loss, knowledge routing impact). 2) Insufficient experimental rigor and analysis, including missing comparisons (e.g., to other knowledge-enhanced methods, other fine-tuning techniques), unvalidated synthetic data, incomplete results, and inadequate investigation into performance degradation and model scale effects. 3) Limited novelty and generalizability, with the approach seen as a standard pipeline applied to a specific benchmark, lacking evidence of generalization to unseen objects or broader contributions to multimodal reasoning.

**Reviewer Concerns:**

The reviewer concerns are partially addressed by the author's rebuttal and the feedback from reviewers.  For example, the authors' rebuttal has addressed some specific clarifications regarding experimental settings (e.g., missing baselines, configuration details). However, some main concerns remain largely outstanding, e.g., novelty of SFT+DPO routine, trade-off in accuracy between normal and abnormal cases, and performance trade-offs. These issues are central to the paper's validity and contribution.

**Reviewer Scores:**

After rebuttal, two reviewers acknowledge the merits of the paper, and recommends acceptance, another two reviewers keeps their original rejection scores without positive feedbacks. After carefully reviewing the returned feedbacks and discussions, AC deems that this work requires significant improvements to meet the bar for publication, and agrees that the current version is not acceptable for publication.

---

### Decision · Program_Chairs · 2026-01-26

Reject